# Balance, Imbalance, and Rebalance: Understanding Robust Overfitting from a Minimax Game Perspective

**Yifei Wang**[1]*
yifei_wang@pku.edu.cn

**Liangchen Li**[2]*
ll673@cam.ac.uk

**Jiansheng Yang**[1]
yjs@math.pku.edu.cn

**Zhouchen Lin**[3, 4, 5]†
zlin@pku.edu.cn

**Yisen Wang**[3, 4]†
yisen.wang@pku.edu.cn

[1] School of Mathematical Sciences, Peking University
[2] Department of Engineering, University of Cambridge
[3] National Key Lab of General Artificial Intelligence,
School of Intelligence Science and Technology, Peking University
[4] Institute for Artificial Intelligence, Peking University
[5] Peng Cheng Laboratory

## Abstract

Adversarial Training (AT) has become arguably the state-of-the-art algorithm for extracting robust features. However, researchers recently notice that AT suffers from severe robust overfitting problems, particularly after learning rate (LR) decay. In this paper, we explain this phenomenon by viewing adversarial training as a dynamic minimax game between the model trainer and the attacker. Specifically, we analyze how LR decay breaks the balance between the minimax game by empowering the trainer with a stronger memorization ability, and show such imbalance induces robust overfitting as a result of memorizing non-robust features. We validate this understanding with extensive experiments, and provide a holistic view of robust overfitting from the dynamics of both the two game players. This understanding further inspires us to alleviate robust overfitting by rebalancing the two players by either regularizing the trainer's capacity or improving the attack strength. Experiments show that the proposed ReBalanced Adversarial Training (ReBAT) can attain good robustness and does not suffer from robust overfitting even after very long training. Code is available at https://github.com/PKU-ML/ReBAT.

## 1 Introduction

Overfitting seems to have become a history in the deep learning era. Contrary to the traditional belief in statistical learning theory that large hypothesis class will lead to overfitting, Zhang et al. [58] note that DNNs have good generalization ability on test data even if they are capable of memorizing random training labels. Nowadays, large-scale training often does not require early stopping, and it is observed that longer training simply brings better generalization [19]. However, researchers recently notice that in Adversarial Training (AT), overfitting is still a severe issue on both small and large scale data and models [38]. As the most effective defence against adversarial perturbation [2], AT solves a minimax optimization problem with training data $\mathcal{D}_{\text{train}}$ and model $f_\theta$ [15, 30, 49]:

$$\min_\theta \mathbb{E}_{\bar{x}, y \sim \mathcal{D}_{\text{train}}} \max_{x \in \mathcal{E}_p(\bar{x})} \ell_{\text{CE}}(f_\theta(x), y), \tag{1}$$

where $\ell_{\text{CE}}$ denotes the cross entropy (CE) loss function, $\mathcal{E}_p(\bar{x}) = \{x \mid \|x - \bar{x}\|_p \leq \varepsilon\}$ denotes the $\ell_p$-norm ball with radius $\varepsilon$. However, contrary to standard training (ST), AT suffers severely from

---

*Equal Contribution.

†Corresponding Authors: Yisen Wang (yisen.wang@pku.edu.cn) and Zhouchen Lin (zlin@pku.edu.cn)

37th Conference on Neural Information Processing Systems (NeurIPS 2023).

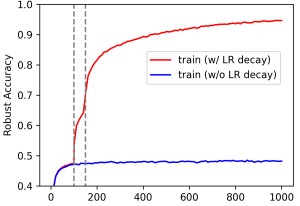

(a) training robustness during adversarial training

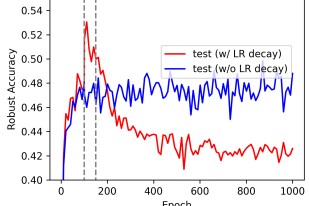

(b) test robustness during adversarial training

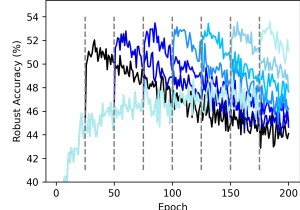

(c) test robustness when decay at different time

Figure 1: (a, b) Training/test robust accuracy on CIFAR-10 using vanilla PGD-AT [30] w/ and w/o LR decay. (c) RO happens whenever LR decays.

robust overfitting (RO): after a particular point (*e.g.,* learning rate decay), its training robustness will keep increasing (Figure 1a, red line) while its test robustness decreases dramatically (Figure 1b, red line). This abnormal behavior of AT has attracted much interest. Previous works find that the AT loss landscape becomes much sharper during RO [42, 8, 53], but we are still unclear about what causes the sharp landscape. Some researchers try to explain RO through known ST phenomena, such as, random label memorization [12] and double descent [11]. However, these theories cannot explain why only AT overfits while ST hardly does.

Instead, we argue that the reason of RO should lie exactly in the *difference* between AT and ST. Sharing similar training recipes and loss functions, their key difference is that AT adopts a *bilevel minimax objective*, compared to the min-only ST objective. We can think of AT as a minimax game between two players (Eq. 1): the model trainer $\mathcal{T}$ (minimizing loss via updating $\theta$) and the attacker $\mathcal{A}$ (maximizing loss via updating $x$), both adopting first-order algorithms in practice. With a proper configuration of both players, AT can strike a balance during which model robustness hinges on a constant level. In comparison, we also notice that when one player (typically the trainer) changes its original strategy (*e.g.,* decaying learning rate), the original balance of the game is broken and RO also occurs subsequently. We further verify this observation through a controlled experiment. As shown in Figure 1c, if we do not perform LR decay, RO will not occur till the end; but whenever we perform LR decay, RO will immediately happen. This provides strong evidence showing that breaking the balance between minimax layers can cause robust overfitting.

To gain further insights into this balance-imbalance process, in Section 3, we provide a generic explanation for RO from the perspective of feature learning. Specifically, when the balance breaks after the LR decay, the trainer with a smaller LR is endowed with a stronger local fitting ability capable of memorizing adversarial examples such that the attacker can no longer succeed in attack. However, during this process, the model learns false mapping of non-robust features contained in the training adversarial examples, which opens shortcuts for the test-time attacker and induces large test robust error. Accordingly, we design a series of experiments that align well with our analysis.

The minimax game perspective also indicates a simple fix to RO, that is to adjust the ability of the two minimax players to a new balance by modifying their strategies, namely, *rebalancing*. In this work, we explore three new strategies for rebalancing the game after LR decay: bootstrapping objective, smaller LR decay factor, and stronger attacker. The former two restrict the local fitting ability of the trainer while the third enhances the ability of the attacker. We show that all the three strategies help mitigate robust overfitting to a large extent. Based on these insights, we propose ReBalanced Adversarial Training (ReBAT) that achieve superior robustness with a neglectable (if any) degree of robust overfitting, even if we train it further to 500 epochs. Meanwhile, ReBAT attains superior robustness (both best-epoch and final-epoch) on benchmark datasets including CIFAR-10, CIFAR-100, and Tiny-ImageNet. This suggests that with appropriately balanced training strategies, AT could also enjoy similar "train longer, generalize better" property like ST. To summarize, our main contributions are:

- We propose to understand adversarial training (AT) as a dynamic minimax game between two actual players: the model trainer $\mathcal{T}$ and the attacker $\mathcal{A}$. We characterize the balance status of the minimax game from the perspective of non-robust features.

- We show robust overfitting is caused by the imbalance between two minimax players induced by the LR decay, and explain this process as the false memorization of non-robust

features. We design a series of experiments to verify this explanation and provide a holistic characterization of robust overfitting from the views of both players.

- From our dynamic perspective, we propose three effective approaches that mitigate robust overfitting by re-striking a balance between the model trainer and the attacker. Experiments show that our proposed ReBAT method largely alleviates robust overfitting while attaining good robustness on different network architectures and benchmark datasets.

## 2 Preliminaries

**Adversarial Attack.** Given an image classifier $f$, adversarial attack [43] is proposed to perturb the image by an imperceptible noise such that the image is misclassified by $f$. A well-known white-box attacker $\mathcal{A}$ is PGD (Projected Gradient Descent) [30], which uses multi-step gradient ascent steps to maximize the cross entropy loss while projecting intermediate steps to the feasible region:

$$\tilde{x}_{k+1} = x_k + \alpha \nabla_{x_k} \ell_{\text{CE}}(f_\theta(x_k), y), \ x_{k+1} = \mathcal{P}_{\mathcal{E}_p(\bar{x})}(\tilde{x}_{k+1}), \ k = 0, \dots, K-1. \tag{2}$$

where $x_0 = \bar{x} + \varepsilon, \varepsilon \sim U[0, \varepsilon]$, and $\mathcal{E}_p(\bar{x}) = \{x | x \in [0, 1]^d, \|x - \bar{x}\| \le \varepsilon\}$ is the feasible region.

**Adversarial Training.** To counter misclassification caused by the attacker, adversarial training (AT) [43, 15, 30, 38, 50, 3] aims to build a robust classifier through the minimax objective in Equation 1. In practice, AT iterates between two steps: 1) solve the inner-loop maximization *w.r.t.* input $x$ using a few (*e.g.,* 10) PGD steps (Eq. 2) to generate a minibatch of adversarial examples $\{\hat{x}\}$, and then 2) solve the outer-loop minimization *w.r.t.* NN parameters $\theta$ with SGD as the trainer $\mathcal{T}$. Therefore, AT can be seen as a minimax game between two players (specifically, two first-order optimization algorithms), the attacker $\mathcal{A}$ and the trainer $\mathcal{T}$, with the objective function $\ell_{\text{CE}}(f_\theta(\hat{x}), y)$.

**Robust and Non-robust Features.** The arguably most prevailing understanding of adversarial training is the framework proposed by Ilyas et al. [21], where they regard training data as a composition of robust and non-robust features that are both useful for classification. Consider a binary classification task, a feature $f : \mathcal{X} \to \mathbb{R}$ is categorized as 1) a **useful feature** if $\mathbb{E}_{\bar{x},y}[y \cdot f(x)] \ge \rho$; 2) a **robust feature** if it is useful for $\rho > 0$ and $\mathbb{E}_{\bar{x},y} \inf_{x \in \mathcal{E}_p(\bar{x})}[y \cdot f(x)] \ge \gamma$; 3) a **non-robust feature** if it is useful for $\rho > 0$ and not robust for any $\gamma \ge 0$. We can also choose a threshold $\gamma$ to determine its belonging class since there is no clear dividing line between robust and non-robust features in practice. According to Ilyas et al. [21], standard training (ST) adopts non-robust features for better accuracy, which also leads to adversarial vulnerability. Instead, AT only uses robust features and becomes resistant to adversarial attack. Our work also adopts the notion of non-robust features. But compared with their framework which mainly considers the ideal, static solution to the minimax problem (a learned robust classifier), our work zooms into the dynamic interplay of the two actual players (how a classifier learns the robust & non-robust features in AT) to understand RO.

## 3 A Minimax Game Perspective on Robust Overfitting

In this section, we propose a minimax game perspective to understand the cause of robust overfitting. We first show the minimax game in Equation 1 achieves a balance when the trainer $\mathcal{T}$ is not capable of memorizing the non-robust features, and then demonstrate through several verification experiments that the break of this balance, which leads to memorization of the non-robust features, actually induces shortcuts for test-time adversarial attack that results in RO.

### 3.1 AT Strikes a Balance when Non-robust Features Cannot be Memorized

First, we look at the balanced status of adversarial training. From Figure 1a (blue line), we observe that without LR decay, the two players can maintain a constant robustness level throughout training. Inside the equilibrium, the attacker $\mathcal{A}$ injects non-robust features from a different class $y'$ to $x$ in order to misclassify an input $(x, y)$ [21] [3], while the trainer can only correctly classify a part of them using robust features. In other words, the game can strike a balance when there is a limit of the trainer's fitting ability that after a certain robustness level, it can no longer improve robustness by fitting the non-robust features that the attacker can modify (*i.e.,* non-robust features are not memorizable).

To verify this point, we collect misclassified adversarial examples (containing non-robust features *w.r.t.* $y'$ if misclassified to $y'$) on an early checkpoint (60th epoch) and evaluate their accuracy on

---

[3]We refer the reader to Appendix B for simple a verification and to Fowl et al. [14] for a further discussion.

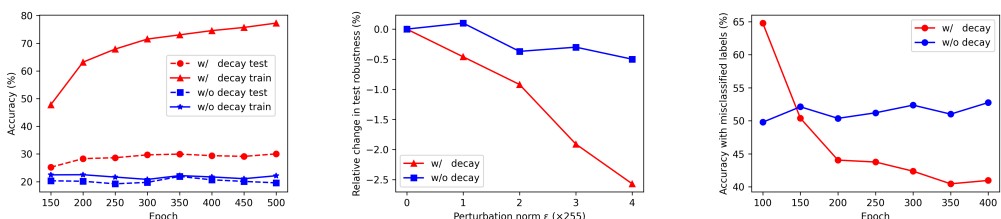

(a) non-robust feature memorization    (b) influence of stronger non-robust features (larger $\varepsilon$) on test robustness    (c) target-class non-robust features

Figure 2: (a) Almost non-generalizable non-robust features in training data is memorized only after LR decay. (b) With LR decay, injecting stronger non-robust features (larger $\varepsilon$) induces severer degradation in test robustness, while the degradation is less obvious w/o LR decay. (c) After LR decay, test adversarial examples contains less and less target-class non-robust features.

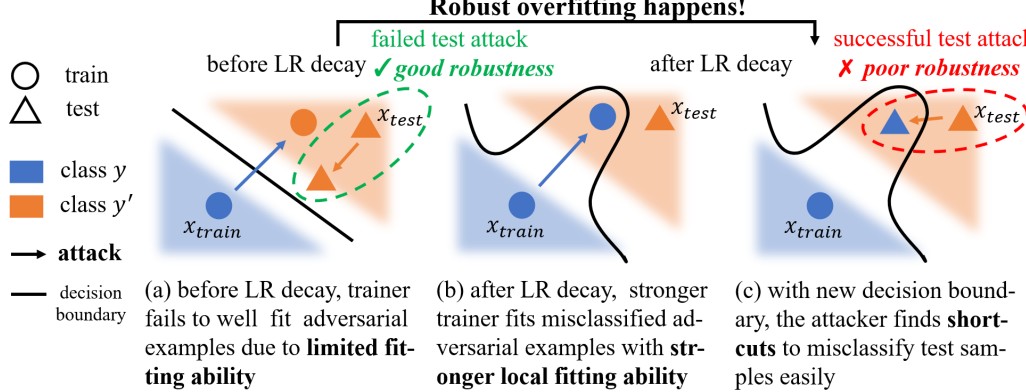

Figure 3: An illustration example of the rise of robust overfitting. Here, the attack (from the clean example to the adversarial example) follows the direction to the nearest decision boundary.

a later checkpoint ($\geq$150th epoch) *w.r.t.* their true labels $y$ (detailed settings in Appendix C.2). If the adversarial non-robust features can be memorized, we would expect high accuracy on these adversarial examples on the later checkpoints. However, in practice, the blue solid line in Figure 2a shows the accuracy is still relatively low ($< 25\%$), suggesting that these non-robust features are almost not memorized under the same training configuration without LR decay.

## 3.2   Balance $\rightarrow$ Imbalance: Robust Overfitting as a Result of Imbalanced Minimax Game

Next, we look at how LR decay breaks the balance of the minimax game. As pointed out by the theoretical work [28], small LR favors easy-to-generalize and hard-to-fit patterns, which correspond to non-robust features in AT [21]. Thus, with a smaller LR, the trainer becomes more capable of memorizing these features by drawing more complex decision boundaries in a small local region. Indeed, we observe a significant increase in training robust accuracy in Figure 1a. Moreover, we can see from the red line in Figure 2a that most of the pre-LR-decay adversarial examples (with adversarial non-robust features) are correctly classified after LR decay, indicating stronger memorization of non-robust features. Therefore, we claim that *LR decay induces an imbalance between the attacker $\mathcal{A}$ and the trainer $\mathcal{T}$*: the boosted local fitting ability of $\mathcal{T}$ overpowers the attack ability of $\mathcal{A}$ that it is capable of memorizing the adversarial non-robust features that $\mathcal{A}$ can leverage.

Meanwhile, as the red dashed line in Figure 2a implies, it turns out that these memorized non-robust features in training data do not generalize to improve test robustness, resulting in a large robust generalization gap (see Figure 1). Next, we further reveal how such memorization harms test robustness by inducing robust overfitting.

### 3.2.1   How Imbalance Leads to Robust Overfitting

Under the imbalanced minimax game, the stronger trainer pushes up training robustness by memorizing the non-robust features in the training adversarial examples. However, the trainer's (overly) strong fitting ability actually learns false mappings of non-robust features which induce shortcuts for

test-time adversarial attack that lead to robustness degradation. Below, we explain how this happens during adversarial training, and an (over-simplified) illustration is shown in Figure 3.

**1) Model Learns False Mapping of Non-robust Training Features after Decay.** Consider a training adversarial example $x'$ from class $y$ that is misclassified to class $y' \neq y$ before LR decay. According to [21], the non-robust feature $g$ of $x'$ belongs to class $y'$, *i.e.,* $g \in \mathcal{F}_{y'}$. After LR decay, the trainer $\mathcal{T}$ becomes more capable of memorizing with a smaller LR and it successfully learns to map this adversarial example $x'$ to its correct label $y$. Accordingly, as we argued above, the non-robust feature $g \in \mathcal{F}_{y'}$ in $x$ is also (falsely) mapped to class $y$, *i.e.,* $\varphi(g) = y$.

**2) False Non-robust Mapping Opens Shortcuts for Test-time Adversarial Attack.** Now we consider a clean test sample $\hat{x}$ whose true label is $y'$. Before LR decay, in order to misclassify it, the attacker $\mathcal{A}$ has to add non-robust features *from a different class* , say $g' \in \mathcal{F}_{\hat{y}'}, \hat{y}' \neq y'$. However, after LR decay, because of the existence of false non-robust mapping $\varphi$, the attacker $\mathcal{A}$ can simply add the non-robust feature from the same class, *i.e.,* $g \in \mathcal{F}_{y'}$ such that $\varphi(g) = y, y \neq y'$ to misclassify $\hat{x}$ to $y$. In other words, the injected non-robust features do not even have to come from a different class to misclassify $\hat{x}$, which are much easier for the attacker to find when starting from $\hat{x}$ (compared with non-robust features $g'$ from other classes). Therefore, the attacker will have a larger attack success rate with the help of these *falsely memorized non-robust features*, as they create easy-to-find shortcuts for test-time adversarial attack.

**Remark.** To be rigorous, we give a formal definition for these non-robust features. Denote the hypothesis class of features that the model can robustly fit on training data under configuration $t$ as

$$\mathcal{H}_t = \Big\{ f_\theta : \mathcal{X} \to \mathbb{R} \mid \mathbb{E}_{\bar{x}, y \sim \mathcal{D}_{train}} \inf_{x \in \mathcal{E}_p(\bar{x})} [y \cdot f_\theta(x)] > 0,$$

$$\text{where } f_\theta \text{ is robust on training data, and } \theta \in \Theta \text{ is attainable under the configuration } t \Big\}. \tag{3}$$

We say a non-robust feature $f$ is falsely memorized if $f \notin \mathcal{H}_{\text{before}}, f \in \mathcal{H}_{\text{after}}$, where $\mathcal{H}_{\text{before}}, \mathcal{H}_{\text{after}}$ refer to the hypothesis classes before and after LR decay, respectively. In other words, the non-robust $f$ is falsely memorized by the model (as a robust feature) under the after-LR-decay trainer. This mismatch could happen because the trainer only sees training examples and it can overfit them under an imbalanced minimax game. Since this feature is essentially a non-robust feature (in the population sense), this feature still behaves non-robust on test data and introduce shortcuts to test-time attack, as we elaborate above.

### 3.2.2 Verification

To further validate our explanation above, we design a series of experiments regarding the influence of memorization of the adversarial non-robust features after LR decay (Verification I & II). Guided by our explanation, we also observe some new intriguing phenomena of RO, which also help verify our minimax perspective (Verification III & IV). Please refer to Appendix C.2 for more details.

**Verification I: More Non-robust Features, Worse Robustness.** To study the effect of non-robust features, we deliberately add non-robust features of different strengths to the clean data (a larger $\varepsilon$ indicates stronger non-robust features) and perform AT on these synthetic datasets. As shown in Figure 2b, we can see that 1) with LR decay, stronger non-robust features induce severer test robustness degradation, and 2) it is much less severe without LR decay. This directly shows that the memorization of non-robust features induced by LR decay indeed hurts test robustness a lot.

**Verification II: Vanishing Target-class[4] Features in Test Adversarial Examples.** In our example above, the added non-robust feature $g$ comes from the same class $y'$ as $\hat{x}$, so the resulting adversarial example $\hat{x}'$ actually contains little features of class $y$, even if misclassified to $y$. Figure 2c shows that the test adversarial examples generated after LR decay indeed have less and less target-class features as RO becomes more and more severe, which proves the shortcuts indeed exist to make test-time adversarial attack easier.

**Verification III: Bilateral Class Correlation.** Our theory suggests that if RO happens, the training misclassification from class $y \to y'$ (before decay) will induce an increase in test misclassification from class $y' \to y$ (after decay), as the $y \to y'$ false non-robust mappings create $y' \to y$ shortcuts. We verify this in Figure 4a, in which we observe that this bilateral correlation indeed consistently increases after LR decay, while remaining very low (nearly no correlation) without decay.

---

[4]By target-class we mean the class to which an adversarial example is misclassified.

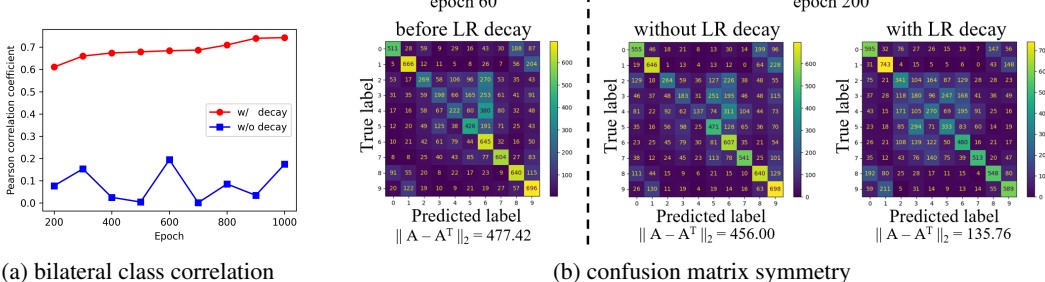

(a) bilateral class correlation             (b) confusion matrix symmetry

Figure 4: (a) Increasingly strong correlation between training-time $y \to y'$ misclassification and test-time $y' \to y$ misclassification changes (due to RO). (b) Test-time confusion matrix of adversarial examples becomes symmetric with LR decay.

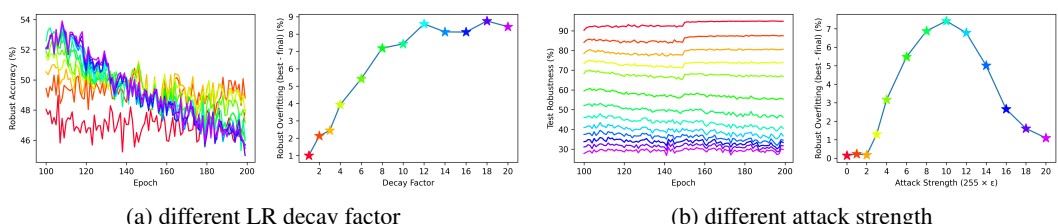

(a) different LR decay factor             (b) different attack strength

Figure 5: (a) RO becomes increasingly severe as larger LR decay factor is adopted. (b) Varying the attack strength from extremely weak to extremely strong, the degree of RO first increases and then decreases after a turning point around $\varepsilon = 10/255$.

**Verification IV: Symmetrized Confusion Matrix.** As a result of the bilateral correlation, we can deduce a very interesting phenomenon that the $y \to y'$ and $y' \to y$ misclassification will have more similar error rates, which means that the confusion matrix of test robustness will become more symmetric. Indeed, as shown in Figure 4b, the confusion matrix (denoted as $A$) becomes much more symmetric (smaller distance between $A$ and $A^\top$) after LR decay than without LR decay.

### 3.3 A Holistic View of Robust Overfitting from the Minimax Game Perspective

Based on our understanding of RO through the lens of the minimax game, we give a holistic view of RO, revealing the influence of both the trainer and the attacker.

**Trainer.** We first fix the attacker strength to be PGD-10 attack with $\varepsilon = 8/255$ and change the LR decay factor $d$ from 1 (w/o LR decay) to 20 at epoch 100, and we observe that stronger LR decay induces a monotonically increasing degree of RO (Figure 5a). This agrees well with our explanation that smaller LR induces a severer imbalance and can memorize more non-robust features that eventually harm test robustness.

**Attacker.** We then fix the training configuration and vary the attack strength from PGD-0 with $\varepsilon = 0/255$ (equivalent to standard training (ST)) to PGD-25 with $\varepsilon = 20/255$. Different from previous beliefs that RO will become severer with stronger attack [54], we find an intriguing inverted U curve in Figure 5b: the degree of RO firstly rises with stronger attackers, peaks at around $\varepsilon = 10/255$, and decreases under even stronger attackers ($\varepsilon > 10/255$). Our theory can naturally explain this phenomenon: 1) when using a weak attacker (small $\varepsilon$), training robustness is very high and there are only a few non-robust features. Thus memorizing them only leads to slight or no RO; 2) a very strong attacker (e.g., $\varepsilon > 10/255$) can also counter the strong fitting ability of the post-decay trainer, which also prevents the memorization of non-robust features and alleviates RO.

In summary, the conditions for RO are two folds: 1) the attacker $\mathcal{A}$ should be strong enough to generate enough non-robust features (which explains why AT overfits while ST does not); and 2) the trainer $\mathcal{T}$ should be stronger than $\mathcal{A}$ to memorize the non-robust features (which explains how LR decay induces RO). This understanding also suggests that we can alleviate RO by restoring the balance between the minimax players, as we explore in the next section.

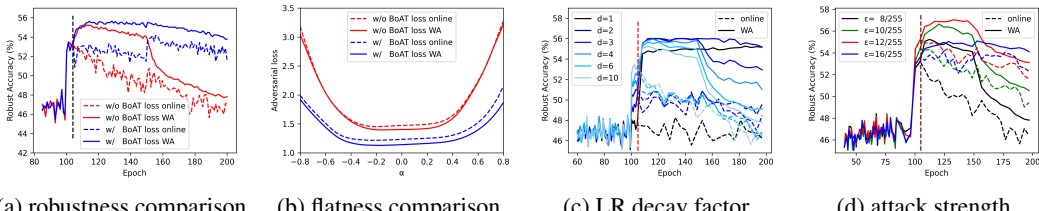

| (a) robustness comparison | (b) flatness comparison | (c) LR decay factor | (d) attack strength |

Figure 6: Mitigating RO from three different techniques. (a, b) The purposed BoAT loss successfully boosts model robustness and mitigates RO by bootstrapping loss landscape flatness between online and WA model. (c) Small LR decay factor leads to better best-epoch robustness of WA model with only slight RO. (d) Stronger training attacker also helps mitigate RO and boost robustness.

# 4 Mitigating Robust Overfitting by Rebalacing the Minimax Game

Based on our understanding of RO from the imbalanced minimax game perspective, we further investigate how to alleviate RO by restoring balance. In particular, we can either weaken the trainer's fitting ability (Sections 4.1), or strengthen the attacker's attacking ability (Section 4.2). We observe that strategies from both directions can prevent the model trainer $\mathcal{T}$ from fitting non-robust features too quickly and too adequately, thus can mitigate RO.

## 4.1 Trainer Regularization

From the trainer's side, one approach to rebalance the game is to regularize the trainer's local fitting ability after LR decay, *e.g.,* by enforcing better landscape flatness. In this way, we could prevent the learner from drawing complex decision boundaries and memorizing non-robust features. Among existing approaches targeting at this, weight averaging (WA) [22] taking the (moving) average of history model weights is effective and efficient that it brings neglectable training overhead, so we focus on WA in this work. Along the online model $f_\theta$ that is adversarially trained, we also maintain a WA model $f_\varphi$ that is an exponential moving average (EMA) of the online parameters $\varphi \leftarrow \gamma \cdot \varphi + (1 - \gamma) \cdot \theta$, where $\gamma \in [0, 1]$ is the decay rate.

**Bootstrapping.** We notice that when the online model $f_\theta$ deteriorates, the WA model, although flatter [22], will eventually deteriorate as a moving average of $f_\theta$ (Figure 6a). Inspired from the latent bootstrap mechanism in self-supervised learning [16], we align the predictions of $f_\varphi$ and $f_\theta$ to improve the flatness of the online model $f_\theta$ simutaneously. Specifically, given an adversarial example $x$ generated by PGD attack using vanilla AT loss (Eq. 1), we have

$$\ell_{\mathrm{BoAT}}(x, y; \theta) = \ell_{\mathrm{CE}}(f_\theta(x), y) + \lambda \cdot \mathrm{KL}\left(f_\theta(x) \| f_\varphi(x)\right), \qquad (4)$$

where $\lambda$ is a coefficient balancing the CE loss (for better training robustness) and the KL regularization term (for better landscape flatness). In one direction, the term provides more fine-grained supervision from the flatter WA model that helps avoid overfitting to the training labels [33, 8], leading to better loss landscape flatness and robustness of the online model; in return, a better online model will further improve the flatness of the WA model. Since this regularization encourages loss landscape flatness of both $f_\varphi$ and $f_\theta$ in a bootstrapped fashion, we name it Bootstrapped Adversarial Training (BoAT) loss.

We further verify the effectiveness of the proposed BoAT loss in Figure 6a, in which both the online model and the WA model suffer significantly less from RO and achieve higher best-epoch robustness compared with using the vanilla AT loss (see Appendix D.1 for details). Moreover, we compare their 1-D loss landscapes [25] at the last checkpoints, and Figure 6b demonstrates that both the online model and the WA model trained with our BoAT loss enjoy much better flatness, which implies weak memorization of non-robust features under the rebalanced game that explains the alleviation in RO.

**Smaller LR Decay Factor.** As studied in Section 3, LR decay has a dramatic influence on RO and reducing the LR decay factor can largely ease RO (Figure 5a). Therefore, we also adopt a smaller LR decay factor (together with the BoAT loss) to further regularize the trainer's fitting ability. To demonstrate its effectiveness, even when we go back to the vanilla AT loss, Figure 6c shows the WA model obtained with a very small decay rate $d = 2$ not only has very slight RO but also attains better best test robustness than $d = 10$. We also remark how the WA model benefits from continuous supply of good individuals, which couldn't be guaranteed under strong RO, also throws new meaningful light on our effort in mitigating RO.

Table 1: Comparing our method with several training methods on CIFAR-10 under the perturbation norm $\varepsilon_\infty = 8/255$ based on the PreActResNet-18 and WideResNet-34-10 architectures.

| | PreActResNet-18 | | | | | | WideResNet-34-10 | | | | | |
|---|---|---|---|---|---|---|---|---|---|---|---|---|
| Method | Natural | | PGD-20 | | AutoAttack | | Natural | | PGD-20 | | AutoAttack | |
| | best | final | best | final | best | final | best | final | best | final | best | final |
| PGD-AT | 81.61 | 84.67 | 52.51 | 46.20 | 47.51 | 42.31 | 86.38 | 87.04 | 56.62 | 48.54 | 51.94 | 45.87 |
| TRADES | 80.45 | 82.89 | 52.32 | 49.39 | 48.09 | 46.40 | 84.23 | 84.89 | 56.02 | 47.38 | 52.76 | 45.62 |
| WA | 83.50 | **84.94** | 55.05 | 47.60 | 49.89 | 43.83 | **87.66** | 87.12 | 56.62 | 49.12 | 52.65 | 46.34 |
| KD+SWA | **84.06** | 84.48 | 53.70 | 53.68 | 49.82 | 49.37 | 87.45 | **88.21** | 56.29 | 55.76 | 53.59 | 53.55 |
| PGD-AT+TE | 82.04 | 82.59 | 54.98 | 53.54 | 50.12 | 49.09 | 85.97 | 85.77 | 56.42 | 53.40 | 52.88 | 50.56 |
| AWP | 81.11 | 80.62 | 55.60 | 55.03 | 50.09 | 49.85 | 85.63 | 85.61 | 58.95 | 59.05 | 53.32 | 53.38 |
| MLCAT$_{WP}$ | - | - | **58.48** | **57.65** | 50.70 | 50.32 | - | - | **64.73** | **63.94** | 54.65 | 54.56 |
| **ReBAT** | 81.86 | 81.91 | 56.36 | 56.12 | 51.13 | 51.22 | 85.25 | 85.52 | 59.59 | 59.29 | 54.78 | 54.80 |
| **ReBAT++** | 78.71 | 78.85 | 56.68 | 56.70 | **51.49** | **51.39** | 81.65 | 81.59 | 59.81 | 59.88 | 54.80 | 54.91 |
| WA+CutMix | 80.24 | 80.30 | 56.02 | 55.95 | 49.67 | 49.59 | 85.08 | 87.95 | 60.41 | 59.36 | 55.11 | 53.60 |
| **ReBAT+CutMix** | 79.02 | 78.95 | 56.15 | 56.16 | 50.18 | 50.22 | 86.28 | 87.01 | 61.33 | 61.24 | **55.75** | **55.72** |

Table 2: Comparing our method with several training methods on CIFAR-100 and Tiny-ImageNet under the perturbation norm $\varepsilon_\infty = 8/255$ based on the PreActResNet-18 architecture.

| | CIFAR-100 | | | | | | Tiny-ImageNet | | | | | |
|---|---|---|---|---|---|---|---|---|---|---|---|---|
| Method | Natural | | PGD-20 | | AutoAttack | | Natural | | PGD-20 | | AutoAttack | |
| | best | final | best | final | best | final | best | final | best | final | best | final |
| PGD-AT | 55.94 | 56.39 | 29.74 | 22.47 | 24.84 | 19.80 | 45.29 | 48.70 | 21.86 | 17.54 | 17.29 | 14.19 |
| TRADES | 55.04 | 57.09 | 29.17 | 26.36 | 24.21 | 23.06 | 48.32 | 47.59 | 22.25 | 21.14 | 16.55 | 15.99 |
| WA | **57.26** | **58.40** | 30.35 | 23.52 | 25.83 | 20.51 | 49.56 | 49.35 | 24.24 | 19.30 | 19.47 | 15.56 |
| KD+SWA | 57.17 | 58.23 | 29.50 | 29.33 | 25.66 | 25.65 | **50.28** | **50.67** | 24.37 | 24.30 | 19.46 | 19.51 |
| PGD-AT+TE | 56.41 | 57.26 | 30.90 | 29.07 | 25.84 | 24.99 | 46.71 | 50.50 | 22.76 | 19.26 | 18.02 | 16.13 |
| AWP | 54.10 | 54.78 | 30.47 | 30.15 | 25.16 | 25.01 | 43.54 | 43.33 | 23.75 | 23.55 | 18.12 | 18.07 |
| MLCAT$_{WP}$ | - | - | 31.27 | 30.57 | 25.66 | 25.28 | - | - | - | - | - | - |
| **ReBAT** | 56.13 | 56.79 | 32.52 | 32.02 | **27.60** | 27.29 | 48.28 | 47.92 | 24.93 | 24.14 | 20.54 | 19.79 |
| **ReBAT++** | 52.44 | 54.65 | 32.49 | 32.11 | 27.49 | **27.33** | 45.10 | 44.12 | 25.23 | 24.52 | **20.67** | **20.51** |
| WA+CutMix | 56.73 | 57.17 | 31.70 | 31.71 | 26.43 | 26.08 | 46.48 | 46.48 | 24.80 | 24.72 | 18.87 | 18.87 |
| **ReBAT+CutMix** | 56.05 | 56.02 | **32.71** | **32.57** | 27.08 | 27.20 | 46.47 | 46.62 | **25.28** | **25.28** | 19.40 | 19.35 |

## 4.2 Stronger Training-time Attacker

From our minimax game perspective, beside directly restricting the fitting ability of the trainer $\mathcal{T}$, another natural approach is to strengthen the attacker $\mathcal{A}$'s ability to counter it after LR decay to help strike a new balance between the two players. Therefore, we apply four different training perturbation budgets after decay, $\varepsilon = 8/255, 10/255, 12/255, 16/255$ (with fixed step size $\alpha = 2/255$ and increasing PGD steps $k \approx 10 \cdot \varepsilon/(8/255)$), where $\varepsilon = 8/255$ is the default budget used before LR decay and in test-time attack. As can be seen from Figure 6d, the adopted stronger attacker indeed suppress RO for both the online and WA models, thanks to the hard-to-memorize adversarial non-robust features they attached to the clean training data; and again we can see how WA benefits from the alleviation of RO, especially for $\varepsilon = 12/255$ that even hits $> 57\%$ WA model robustness against PGD-20 attack (see also Appendix C.3). Besides, a too strong training attacker may prevent the $\mathcal{T}$ from learning useful robust features (*e.g.*, $\varepsilon = 16/255$) that harms robustness, implying that we should carefully choose the attacker strength while rebalancing the minimax game via this approach.

## 4.3 Overall Approach

Above, we have introduced three effective techniques to mitigate RO from different perspectives: model regularization, learning rate decay, and attacker strength. Since stronger attacker often induces a large decrease in natural accuracy [1] (we additionally study this trade-off in Appendix C.3) and needs more careful balancing in order not to harm robustness as discussed above, we design two

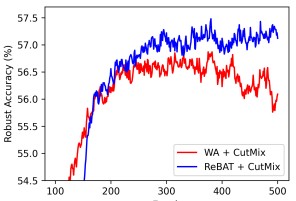
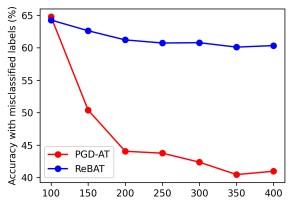
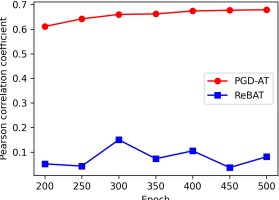

(a) training longer with CutMix     (b) target-class non-robust features     (c) bilateral class correlation

Figure 7: Empirical understandings on ReBAT. (a) ReBAT can benefit from training longer (with CutMix). (b) Compared to PGD-AT, test adversarial examples still have to contain fairly many target-class non-robust features to successfully attack a model trained with ReBAT. (c) ReBAT does not result in strong bilateral class correlation as in PGD-AT.

versions of our final approach: ReBalanced Adversarial Training **(ReBAT)**, which combines the BoAT loss with smaller LR decay factor; and **ReBAT++**, which combines all the three techniques.

## 5 Experiments

In this section, we evaluate the effectiveness of the proposed ReBAT and ReBAT++ on several benchmark datasets using different network architectures. We consider the classification tasks on CIFAR-10, CIFAR-100 [23], and Tiny-ImageNet [10] with the PreActResNet-18 [17] and WideResNet-34-10 [57] architectures. Besides the vanilla PGD-AT [30], we include the following baseline methods for alleviating robust overfitting: WA [22], KD+SWA [8], PGD-AT+TE [12], AWP [53], WA+CutMix [36] and MLCAT$_{WP}$ [54]. During evaluation, we use PGD-20 [30] and AutoAttack (AA) [9] for the adversarial attack methods. We train each model for 200 epochs and record the checkpoint on which the validation set achieves the highest PGD-20 robustness and the final checkpoint for evaluation on the test set. Please refer to Appendix C.4 and C.5 for details and Appendix D for additional results.

### 5.1 Results on Benchmark Datasets

**Performance across Different Networks.** In Table 1, we evaluate ReBAT against previous AT variants on two popular backbone networks: PreActResNet-18 and WideResNet-34-10. We can see that without strong data augmentations like CutMix, ReBAT attains the best robustness on both models and outperforms all baseline methods against AutoAttack (AA), arguably the strongest white-box attacker. Meanwhile, ReBAT almost shows no RO on both PreActResNet-18 (51.13% best and 51.22% final AA robustness), and WideResNet-34-10 (54.78% best and 54.80% final AA robustness). For ReBAT++, it indeed results in even better best and final robustness than ReBAT on both models also with nearly perfect ability in suppressing RO, though at the cost of degradation in natural accuracy as discussed in Section 4.2. The results verify the effectiveness of our proposed method as well as the importance of rebalancing the minimax game.

**Strong Augmentations.** WideResNet-34-10 is more prone to overfit as it has larger capacity, and existing WA+CutMix strategy [36] still overfits by more than 1.5% (55.11% best and 53.60% final AA robustness). In comparison, ReBAT+CutMix achieves 55.75% best robustness (chosen according to PGD) and 55.72% final robustness, showing that this combination indeed alleviates RO very effectively on the relatively large WideResNet-34-10 while maintaining very high robustness. In comparison, CutMix brings little improvement on PreActResNet-18, mainly because this model has lower capacity that still underfits under strong augmentations, and it requires longer training to reach higher performance. Even though, ReBAT+CutMix still outperforms WA+CutMix by 0.63% AA robustness at the conclusion of 200 epochs of training.

**Performance across Different Datasets.** We also conduct experiments on two additional benchmark datasets, CIFAR-100 and Tiny-ImageNet, that are more complex than CIFAR-10. As shown in Table 2, our ReBAT, ReBAT++ and ReBAT+CutMix still achieve the highest best and final robustness, demonstrating its scalability to larger scale datasets. Similarly, CutMix does not help much as it often leads to underfitting with stronger regularization effects within relatively few training epochs.

### 5.2 Empirical Understandings

**ReBAT Can Benefit from Longer Training.** Table 1 suggests that when CutMix is applied to a PreActResNet-18 model, it demands longer training to fully unleash the beast. This motivates us to

examine the effect of longer training, *e.g.,* 500 epochs. As shown in Figure 7a, longer training with WA+CutMix brings little improvement on final robustness (49.59% (200 epochs) *v.s.* 49.70% (500 epochs) under AA). Instead, ReBAT+CutMix achieves higher best robustness and the robustness keeps improving along training. In particular, the additional 300 epochs improve the final AA robustness of ReBAT+CutMix from 50.22% to 51.32%. This provides strong evidence on the effectiveness of ReBAT against RO and also shows that combining ReBAT with strong data augmentations can benefit from long training as in standard training.

**Verification on Robust Overfitting Experiments.** Beyond its success, we empirically study how ReBAT manages to suppress RO through our minimax game perspective. Following the experiments in Section 3.2.2, we compare PGD-AT and ReBAT in terms of 1) target-class non-robust features in test adversarial examples and 2) the strength of bilateral class correlation. We can see that the test adversarial examples still have to contain much target-class non-robust features to be misclassified (Figure 7b) and there is no longer a strong bilateral class correlation (Figure 7c). This confirms that ReBAT indeed successfully prevents the model from learning false mappings of non-robust training features which creates shortcuts for test-time attack.

# 6    Revisting Previous Works from the Minimax Game Perspective

From a dynamic minimax game perspective of AT, we build a generic understanding of RO as a result of memorizing non-robust features under an imbalanced game, and introduce three strategies to rebalance the minimax game to mitigate RO. In fact, we notice that this perspective can also enable a unified understanding that existing approaches to alleviate RO can be perceived as rebalancing techniques developed from the following three dimensions. Additional related works, including discussions on the game-theoretical views of adversarial training, are deferred to Appendix A.

**Data Regularization.** Since RO is induced by non-robust features in the training data, one way is to corrupt those features with various transformations. Since non-robust features often correspond to local and high-frequency patterns in images [21], stronger augmentations like CutMix [56] and IDBH [26] will help mitigate RO [38, 26]. A complementary approach is to correct the *supervision* of the adversarial examples accordingly to avoid learning false label mapping (Section 3.2), *e.g.,* the soft labels adopted in Dong et al. [11], and the distillation-alike methods in KD+SWA [8] and TE [12].

**Training Regularization.** As discussed in Section 3.3, a necessary condition of RO is that the model trainer is stronger than the attacker, so restricting the trainer's ability in fitting non-robust features helps suppress RO. One approach is to directly promote model flatness, *e.g.,* AWP [53], MLCAT$_{\text{WP}}$ [54], EMA [36], SWA [8], AdvLC [27] and our BoAT loss (Section 4.1). Another approach is to adjust the learning rate schedule to avoid dramatic increase in $\mathcal{T}$'s ability, which is previously studied by Rice et al. [38] and Wang et al. [45]. While RO is indeed delayed by some mild LR decay schedules attempted, it still happens as $\mathcal{T}$ eventually becomes strong. Particularly, we show that simply piecewise decay with a small LR decay factor can be significantly helpful for both better robustness and less severe RO.

**Stronger Training Attacker.** Previous attempts in changing attacker strength, including Cai et al. [7], Qin et al. [35] and Wang et al. [49], mainly start training from a *weaker* attacker (*e.g.,* $\varepsilon = 0$), and gradually lift its strength to the standard level ($\varepsilon = 8/255$), which is later pointed out by Pang et al. [33] that these methods are hardly useful when measured by AA robustness. OA-AT [1] introduces a stronger attacker *from the beginning* but aiming to achieve robustness against stronger attacks without much concern in RO. Instead, our method is designed from a minimax game perspective that adopts a *stronger* training attacker ($\varepsilon > 8/255$) *after LR decay* in order to counter RO. Our method shows clear benefits on improving best AA robustness as well as mitigating RO.

# 7    Conclusion

In this paper, we investigate the underlying cause of robust overfitting through a dynamic minimax game perspective of adversarial training. Based on this perspective, we extend the robust feature framework to analyze the training dynamics of AT, and analyze how the change of LR decay induces an imbalance between the two players of AT. Specifically, we have developed a new understanding of robust overfitting through the false memorization of non-robust features, and explored three new approaches to restore the balance between the model trainer and the attacker. Remarkably, the proposed ReBalanced Adversarial Training (ReBAT) have shown that adversarial training can also benefit from long training as long as the robust overfitting issue is resolved.

## Acknowledgements

Yisen Wang was supported by National Key R&D Program of China (2022ZD0160304), National Natural Science Foundation of China (62006153, 62376010, 92370129), Open Research Projects of Zhejiang Lab (No. 2022RC0AB05), Beijing Nova Program (20230484344), and CCF-Baichuan-EB Fund. Zhouchen Lin was supported by the major key project of PCL, China (No. PCL2021A12), the NSF China (No. 62276004), and Qualcomm.

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

# A   Additional Related Work

## A.1   Game-Theoretical View of Adversarial Training

There is a body of literature that analyzes AT from a game theory perspective, see [6, 32, 5, 34, 37, 4]. However, existing theory papers only analyze AT's Nash equilibrium under toy models (e.g., Gaussian features and linear models), and static minimax players. None of the existing literature considers a dynamic game and could explain robust overfitting from a game perspective, particularly in practical AT algorithms, as done in our work. Nevertheless, we find some interesting connections and new insights between them that worth noting.

Under these simplified assumptions, prior works show that although the Nash equilibrium of the AT objective exists and is robust, the current alternating optimization of AT may fail to converge to the Nash equilibrium, see, e.g., a recent work [4]. The key reason is that the trainer can falsely fit the non-robust features [4], which is in a similar spirit to our analysis that robust overfitting is caused by the falsely memorized non-robust features after LR decay. This shows that our explanation is not only in line with the cutting-edge theory of AT, but also further explains robust overfitting from a game perspective for the first time. Besides, we also notice that existing AT game theory papers cannot explain robust overfitting, and from the perspective of our theory, this is possible because they overlook the dynamic change of the AT players. Thus, our understanding of AT from a dynamic game can inspire more in-depth theoretical characterization of AT from the game theory perspective.

As our work mainly focuses on the understanding robust overfitting of practical AT algorithms, and it is generally hard to theoretically analyze the training dynamics of practical DNNs, we leave more theoretical investigations under strong assumptions to future work.

## A.2   Robust and Non-robust Feature View of Adversarial Training

Ilyas et al. [21] firstly propose to understand the existence of adversarial examples from a data feature perspective (introduced in Section 2). Viewing from this framework, adversarial training discards the *useful yet non-robust* features contained in natural images and only utilizes the robust features, which also explains the well-known trade-off between accuracy and robustness [59, 44, 46]. In particular, Ilyas et al. [21] show that robust features are generally aligned with human perception while non-robust features seem random noise. Built upon this observation, researchers explore utilizing adversarial training for representation learning [13, 47] and even image synthesis [40, 48]. This robust feature perspective of adversarial training also inspires various "adversarial-for-good" applications, such as, privacy protection [20, 51].

Despite these fruitful developments from this view, a concurrent work [24] challenges this common understanding by showing that non-robust features are not really useful. In particular, when transferred, non-robust features yield significantly worse performance than robust features under self-supervised learning, contrary to their behaviors under supervised learning. Meanwhile, they also show that robust features alone might not be enough to attain robustness (in practice), since the robust dataset constructed in Ilyas et al. [21] has a false sense of robustness under AutoAttack [9]. Based on these observations, they suggest that non-robust features should be considered as natural backdoor patterns of the supervised learning task alone, instead of universally useful across different learning paradigms. We note that this understanding is actually *not contrary* to our explanation of robust overfitting through the non-robust features, since we only need the usefulness of non-robust features for classification in our discussion. In Section 3.2, we have also mentioned that the falsely memorized non-robust features behave as learning shortcuts, which agrees with the analysis in Li et al. [24].

## A.3   Dynamic Strategies for Adjusting the Trainer and the Attacker in Adversarial Training

In Secion 4, we propose three new strategies to adjust the trainer and the attacker in order to restore the balance of the minimax game. Prior to our work, there existed many proposed strategies to adaptively adjust the trainer or the attacker during adversarial training. For example, Rice et al. [38] show that different schedules of learning rate has a large effect on robust overfitting and the best-epoch robustness, and they propose early stopping to prevent robust overfitting. Wang and Wang [45] instead explore a combination of weight decay and learning and learning rate schedule and obtain better performance. On self-supervised learning, Luo et al. [29] show that a dynamic change

of data augmentation strength can improve robustness significantly. On the other side, Wang et al. [49] propose to dynamically increase the attack steps to attain better convergence, while Mo et al. [31] further apply the dynamic strategy in ViTs. Concurrent to our work, Yu et al. [55] and Wei et al. [52] also empirically find that using a large perturbation budget can mitigate robust overfitting to some extent. In our work, this strategy naturally rises from our minimax game perspective of adversarial training. Compared to sophisticated adaptive designs, our method is simpler to use and delivers better robustness: we only improve the attack strength by a constant after the learning decay.

## B    Additional Experiment: Adversarial Examples (in Adversarial Training) Contain Adversarial Non-robust Features

In Section 3, we collect misclassified adversarial examples on an early checkpoint $A$ and evaluate them on a later checkpoint $B$ with their correct labels to examine the memorization of those adversarial non-robust features. This is because adversarial examples used for AT consist of robust features from the original label $y$ and non-robust features from the misclassified label $\hat{y}$, so $B$ must have memorized those features (non-robust on $A$) to correctly classify them. In this section, we further provide a rigorous discussion on the non-robust features contained in the adversarial examples through an additional experiment.

**Experiment Design.** Recall that Ilyas et al. [21] leverage targeted PGD attack [30] to craft adversarial examples on a standard (non-robust) classifier and relabel them with their target labels to build a non-robust dataset. Finding standard training on it yields good accuracy on clean test data, they prove that those adversarial examples contain non-robust features corresponding to the target labels (Section 3.2 of their paper).

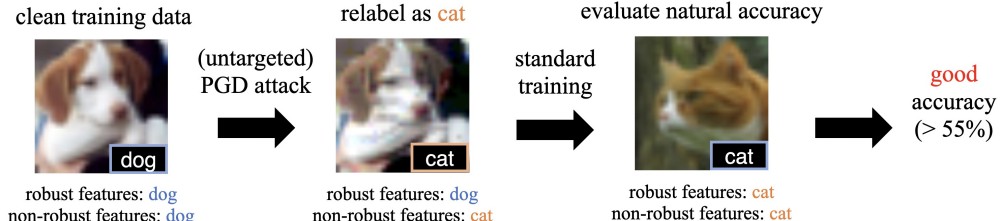

Figure 8: Mining non-robust features from adversarial examples in adversarial training. 1) Craft adversarial examples $\hat{x}_i$ using untargeted PGD attack on a AT checkpoint. 2) Relabel them (only misclassified ones) with misclassified label $\hat{y}_i$ to build a non-robust dataset $\{(\hat{x}_i, \hat{y}_i)\}$. 3) Perform standard training from the checkpoint. 4) Achieve good natural accuracy ($> 55\%$).

However, different from their settings where we know the target labels of the adversarial examples and they are generated on a non-robust classifier, we lay emphasis on mining non-robust features from adversarial examples generated on-the-fly in AT. To this end, we first craft adversarial examples on the AT checkpoint at some epoch using untargeted PGD attack [30] following the real setting of AT, then relabel the *misclassified* ones $\hat{x}_i$ (*e.g.,* a dog) with their misclassified labels $\hat{y}_i \neq y_i$ (*e.g.,* cat) to build a non-robust dataset $\{(\hat{x}_i, \hat{y}_i)\}$. In order to capture non-robust features at exactly the training epoch these adversarial examples are used, we continue to perform standard training directly from the AT checkpoint on the non-robust dataset, and finally evaluate natural accuracy. See Figure 8 for an illustration of our experiment.

In details, we select the AT models at epoch 60 (before LR decay) and epoch 1,000 (after LR decay) to conduct the above experiment. We use untargeted PGD-20 with perturbation norm $\varepsilon_\infty = 16/255$ to craft adversarial examples on those checkpoints from the *training data* of CIFAR-10 [23]. The attack we adopt is a little bit stronger than the attack of PGD-10 with $\varepsilon_\infty = 8/255$ that is commonly used to generate adversarial examples in baseline adversarial training, because the training robust accuracy rises to as high as $94.67\%$ at epoch 1,000 (Figure 1a) and gives less than 2,700 misclassified examples, which is significantly insufficient for further training. Using the stronger attack, we obtain a success rate of $78.38\%$ on the checkpoint at epoch 60 and a success rate of $34.86\%$ on the checkpoint at epoch 1,000. Since the attack success rates are different, we randomly select a same number of misclassified adversarial examples for standard training for a fair comparison. The learning rate is

initially set to 0.1 and decays to 0.01 after 20 epochs for another 10 epochs of fine-tuning. Given that the original checkpoints already have non-trivial natural accuracy, we also add two control groups that train with random labels instead of $\hat{y}$ to exclude the influence that may brought by the original accuracy.

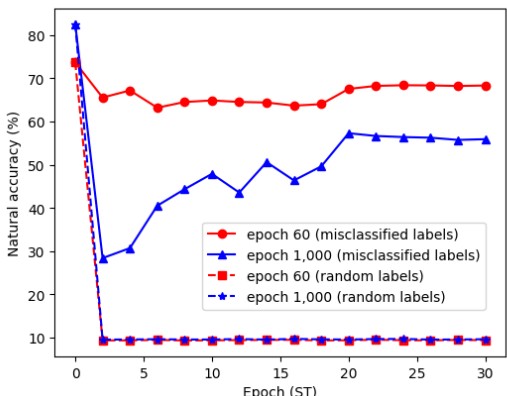

Figure 9: Natural accuracy during standard training. Standard training on the non-robust dataset built from the checkpoint at either epoch 60 or epoch 1,000 converges to fairly good natural accuracy ($> 55\%$). The failure of training with random labels proves that the good accuracy has nothing to do with the original natural accuracy of the AT checkpoint.

**Results.** As shown in Figure 9, we find that standard training on the non-robust dataset $\{(\hat{x}_i, \hat{y}_i)\}$ successfully converges to fairly good accuracy ($> 55\%$) on natural test images, *i.e.,* predicting cats as cats, no matter from which AT checkpoints (either epoch 60 or epoch 1,000) the adversarial examples are crafted. Also, we can see that the non-trivial natural accuracy has nothing to do with the original accuracy of the AT checkpoints, as the accuracy plummets to around 10% (random guessing) as soon as the standard training starts with random labels. This proves that *adversarial examples in adversarial training do contain non-robust features w.r.t. the classes to which they are misclassified*, which strongly corroborates the validity of the experiments in Section 3.

**Discussion on the Influence Brought by Robust Features During Standard Training.** Since untargeted PGD attack cannot be assigned with a target label, we cannot guarantee that the misclassified labels $\hat{y}$ to be uniformly distributed regardless of the original labels (especially for real-world datasets). This implies that the non-robust features are not completely decoupled from robust features, *i.e.,* training on $\{(\hat{x}_i, \hat{y}_i)\}$ may take advantage of $\hat{x}_i$'s robust features from $y_i$ through the correlation between $y_i$ and $\hat{y}_i$. However, we argue that robust features from $y_i$ mingling with non-robust features from $\hat{y}_i$ only increases the difficulty of obtaining a good natural accuracy. This is because learning through the shortcut will only wrongly map the robust features from $y_i$ to $\hat{y}_i$ that never equals to $y_i$, but during the evaluation, each clean test example $x_i$ will always have robust and non-robust features from $y_i$, and such wrong mapping will induce $x_i$ to be misclassified to some $\hat{y}_i$ due to the label of its robust features. As a result, despite the negative influence brought by robust features, we still achieve good natural accuracy at last, which further solidifies our conclusion.

## C   Experiment Details

In this section, we provide more experiment details that are omitted before due to the page limit.

### C.1   Baseline Adversarial Training

In this paper, we mainly consider classification task on CIFAR-10 [23]. The dataset contains 60,000 $32 \times 32$ RGB images from 10 classes. For each class, there are 5,000 images for training and 1,000 images for evaluation. Since we mainly aim to track the training dynamic to verify our understandings in RO instead of formally evaluating an algorithm's performance, we do not hold a validation set in most of our verification experiments and directly train on the full training set.

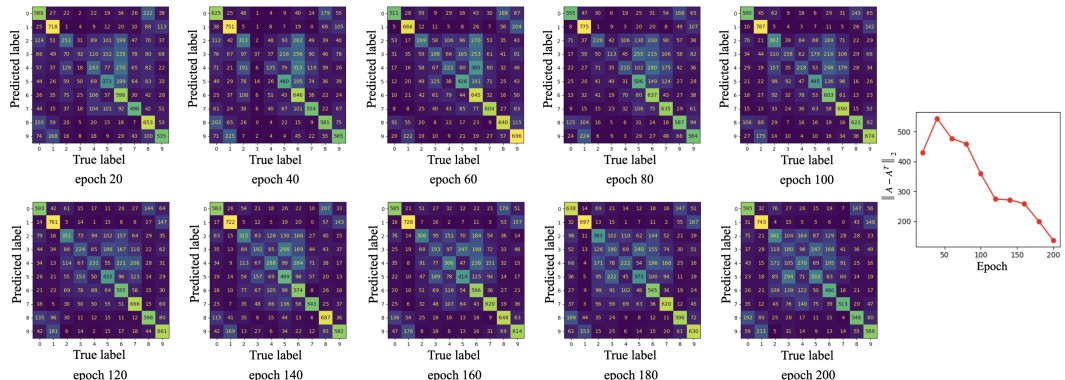

Figure 10: More test-time confusion matrices during the first 200 epochs of the training. After LR decays (the second row), the confusion matrix $A$ immediately becomes symmetric, as the spectral norm $\|A - A^T\|_2$ *w.r.t.* the matrix decreases from $\geq 350$ before epoch 100 to $\leq 150$ after epoch 200.

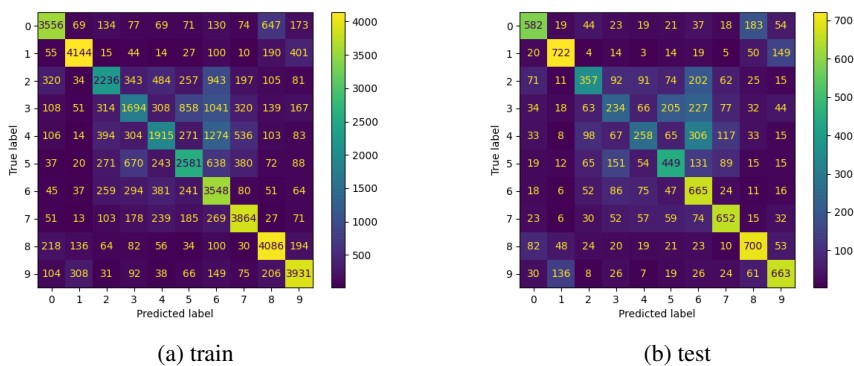

| (a) train | (b) test |
|---|---|

Figure 11: Confusion matrices of the training and the test data at an epoch before robust overfitting starts. They show nearly a same pattern of attacking preference among classes.

For baseline adversarial training, We use PreActResNet-18 [17] model as the classifier. We use PGD-10 attack [30] with step size $\alpha = 2/255$ and perturbation norm $\varepsilon_\infty = 8/255$ to craft adversarial examples on-the-fly. Following the settings in Madry et al. [30], we use SGD optimizer with momentum 0.9, weight decay $5 \times 10^{-4}$ and batch size 128 to train the model for as many as 1,000 epochs. The learning rate (LR) is initially set to be 0.1 and decays to 0.01 at epoch 100 and further decays to 0.001 at epoch 150. For the version without LR decay used for comparison in our paper, we simply keep the LR to be 0.1 during the whole training process.

Each model included in this paper is trained on a single NVIDIA GeForce RTX 3090 GPU. For PGD-AT, it takes about 3d 14h to finish 1,000 epochs of training.

### C.2 Verification Experiments for Our Minimax Game Perspective on Robust Overfitting

**Memorization of Adversarial Non-robust Features After LR Decay.** We craft adversarial examples on a checkpoint before LR decay (60th epoch) and evaluate the *misclassified* ones on a checkpoint after LR decay ($\geq$ 150th epoch) with their correct labels to evaluate the memorization of non-robust features in the training adversarial examples (see Appendix B for detailed discussions) in Section 3.1 and 3.2. Following Appendix B, we adopt PGD-20 attack with perturbation norm $\varepsilon_\infty = 16/255$ to craft adversarial examples which is stronger than the common attack setting we use in PGD-AT. We note that test adversarial examples crafted by a stronger attack indicates stronger extraction of the non-robust features, so they are more indicative of non-robust feature memorization when still correctly classified.

**Verification I: More Non-robust Features, Worse Robustness.** At the beginning of Section 3.2, we create synthetic datasets to demonstrate that memorizing the non-robust training features indeed harms test-time model robustness. To instill non-robust features into the training dataset, we minimize the adversarial loss *w.r.t.* the training data in a way that just like PGD attack, with the only difference that we minimize the adversarial loss instead of maximizing it. Since we only use a very small perturbation norm $\varepsilon_\infty \leq 4/255$, the added features are bound to be non-robust. For a fair comparison, we also perturb the training set with random uniform noise of the same perturbation norm to exclude the influence brought by (slight) data distribution shifts. We continue training from the 100-th baseline AT checkpoint (before LR decay) on each synthetic dataset for 10 epochs, and then evaluate model robustness with clean test data.

**Verification II: Vanishing Target-class Features in Test Adversarial Examples.** This is to say that when RO happens, we expect that test adversarial examples become less informative of the classes to which they are misclassified according to our theory. To verify this, we first craft adversarial examples on a checkpoint $T$ after RO begins, then evaluate the misclassified ones *with their misclassified labels* $\hat{y}$ on the checkpoint saved at epoch 60. As a result, the accuracy reflects how much information (non-robust features) from $\hat{y}$ the adversarial examples have to contain to be misclassified to $\hat{y}$ on $T$. All adversarial examples evaluated in the experiments in Section 3.2.2 are crafted using PGD-10 attack with perturbation norm $\varepsilon_\infty = 8/255$.

**Verification III: Bilateral Class Correlation.** To quantitatively analyze the correlation strength of bilateral misclassification described in Section 3.2.2, we first summarize all $y_i \rightarrow y_j$ misclassification rates into two confusion matrices $P^{\text{train}}, P^{\text{test}} \in \mathbb{R}^{C \times C}$ for the training and test data, respectively. Because we are mainly interested in the effect of the LR decay, we focus on the relative change on the test confusion matrix before and after LR decay, *i.e.,* $\Delta P^{\text{test}} = P^{\text{test}}_{\text{after}} - P^{\text{test}}_{\text{before}}$. According to our theory, for each class pair $(i, j)$, there should be a strong correlation between the training misclassification of $i \rightarrow j$ before LR decay, *i.e.,* $(P^{\text{train}}_{\text{before}})_{ij}$, and the *increase* in test misclassification of $j \rightarrow i$, *i.e.,* $\Delta P^{\text{test}}_{ji}$, as $i \rightarrow j$ training misclassification (may due to intrinsic class bias, as will be further discussed below in details) induces $i \rightarrow j$ false mappings and creates $j \rightarrow i$ shortcuts. To examine their relationship, we plot the two variables $((P^{\text{train}}_{\text{before}})_{ij}, \Delta P^{\text{test}}_{ji})$ and compute their Pearson correlation coefficient $\rho$.

**Verification IV: Symmetrized Confusion Matrix.** In Section 3.2.2, we mention the growing symmetry of the test-time confusion matrix after LR decay as an evidence of the strengthening $y \rightarrow y'$ and $y' \rightarrow y$ correlation. Here we present more confusion matrices during the 200 epochs of the training in Figure 10, and it is very clear that the confusion matrices soon become symmetric after LR decay and RO starts. For a deeper comprehension of this phenomenon, we first visualize the confusion matrices of the training and the test data at an epoch before RO starts in Figure 11. They exhibit nearly a same pattern of attacking preference among classes (*e.g.,* $y \rightarrow y'$) due to the bias rooted in the dataset, *e.g.,* class 6 is intrinsically vulnerable in this case. For the test data, this intrinsic bias wouldn't be wiped out through learning due to the non-generalizability of the memorization of non-robust features in the training data, as discussed at the beginning of Section 3.2 (*i.e.,* $y \rightarrow y'$ bias still holds); and for the training data, this biased feature memorization will open shortcuts for test-time adversarial attack as discussed in Section 3.2 (*i.e.,* $y' \rightarrow y$ begins). Combining both $y \rightarrow y'$ and $y' \rightarrow y$, we arrive at the symmetry of test-time confusion matrix.

**Additional Results: Changing LR Decay Schedule.** Our discussion above is based on the piecewise LR decay schedule, in which the sudden decay of LR most obviously reflects our understandings. Besides, we also explore other LR decay schedules, including Cosine/Linear LR decay, to check whether different LR decay schedules will affect the observations and claims we made in this paper. For each schedule, we train the model for 200 epochs following the settings in Rice et al. [38]. As demonstrated in Figure 12a, we arrive at the same finding as Rice et al. [38] that with Cosine/Linear LR decay schedule, the training still suffers from severe RO after epoch 130. Then, we rerun the empirical verification experiments in Section 3.2.2 and find that under both the two LR decay schedules 1) the test adversarial examples indeed contain less and less target-class non-robust features as the training goes and RO becomes severer and severer (Figure 12b), 2) the bilateral class correlation becomes increasingly strong (Figure 12c) and 3) the confusion matrix indeed becomes symmetric (Figure 13). The results are almost the same as the results achieved when we adopt piecewise LR decay schedule because even though these LR decay schedules are mild, the LR eventually becomes small and makes the trainer $\mathcal{T}$ overly strong to memorize the harmful non-robust features, indicating

that our understandings in the cause of RO is fundamental and regardless of whatever LR decay schedule is used.

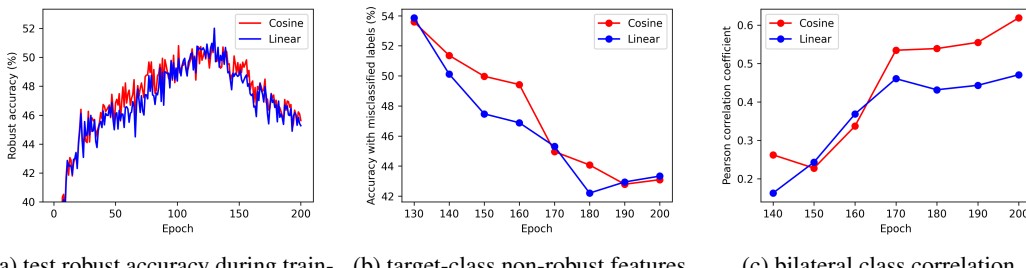

(a) test robust accuracy during train-
ing

(b) target-class non-robust features

(c) bilateral class correlation

Figure 12: Empirical verification of our explanation for robust overfitting when Cosine/Linear LR decay schedule is applied. (a) With Cosine/Linear LR decay schedule, the training still suffers from severe RO. (b) After RO begins, non-robust features in the test data become less and less informative of the classes to which they are misclassified. (c) Increasingly strong correlation between training-time $y \to y'$ misclassification and test-time $y' \to y$ misclassification increase.

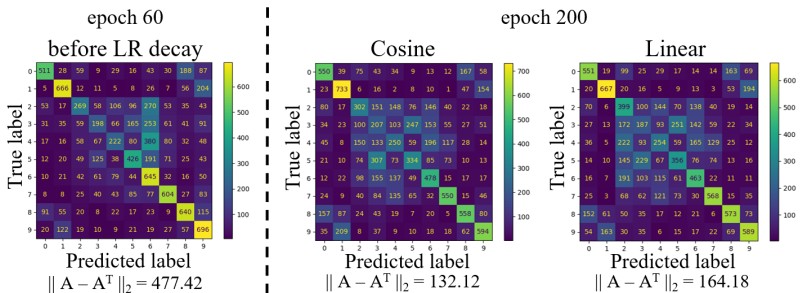

Figure 13: Test-time confusion matrix also becomes symmetric and implies that the bilateral correlation also exists when Cosine/Linear LR decay schedule is applied.

Table 3: Training with stronger attack and evaluating model robustness on CIFAR-10 under the perturbation norm $\varepsilon_{\infty} = 8/255$ based on the PreActResNet-18 architecture.

| Attack Strength | Natural | | | PGD-20 | | | AutoAttack | | |
|---|---|---|---|---|---|---|---|---|---|
| | best | final | diff | best | final | diff | best | final | diff |
| $\varepsilon = 8/255$, PGD-10 (baseline) | **83.50** | **84.94** | **-1.44** | 55.05 | 47.60 | 7.45 | 49.89 | 43.83 | 6.06 |
| $\varepsilon = 10/255$, PGD-12 | 80.66 | 82.48 | -1.82 | 56.63 | 50.63 | 6.00 | 50.89 | 46.13 | 4.76 |
| $\varepsilon = 12/255$, PGD-15 | 78.17 | 80.25 | -2.08 | **57.09** | 53.13 | 3.96 | **50.99** | 47.66 | 3.33 |
| $\varepsilon = 14/255$, PGD-17 | 73.92 | 76.70 | -2.78 | 56.42 | 54.04 | 2.38 | 50.28 | 48.53 | 1.75 |
| $\varepsilon = 16/255$, PGD-20 | 69.51 | 73.11 | -3.60 | 55.09 | **54.27** | **0.82** | 49.58 | **48.53** | **1.05** |

## C.3 Experiments on the Effect of Stronger Training Attacker

In Section 4.2, we point out that using a stronger attacker in AT is able to mitigate RO to some extent by neutralizing the trainer $\mathcal{T}$'s fitting power when it is overly strong. To achieve the results reported in Figure 6d, we craft adversarial examples on-the-fly with more PGD iteration steps when $\varepsilon$ is larger (see Table 3), and further evaluate the best and last robustness of the WA models against PGD-20 and AA. Although RO is only partially mitigated and natural accuracy decreases when a stronger attacker is applied as summarized in Addepalli et al. [1], it may be surprising to find from Table 3 that an attacker of appropriate strength may significantly boost the best WA robustness. This suggests using a stronger attack could potentially be an interesting new path to stronger adversarial defense, and we leave it for future work

## C.4 Detailed Experiment Setup of ReBAT for Mitigating Robust Overfitting

**Datasets.** Beside CIFAR-10, we also include CIFAR-100 [23] and Tiny-ImageNet [10] for evaluation of the effectiveness of ReBAT. CIFAR-100 shares the same training and test images with CIFAR-10, but it classifies them into 100 categories, *i.e.,* 500 training images and 100 test images for each class. Tiny-ImageNet is a subset of ImageNet [10] which contains labeled $64 \times 64$ RGB images from 200 classes. For each class, it includes 500 and 50 images for training and evaluation respectively. Following Rice et al. [38], we hold out 1,000 images from the original CIFAR-10/100 training set, and similarly 2,000 images from the original Tiny-ImageNet training set as validation sets.

**Training Strategy.** For CIFAR-10 and CIFAR-100, we follow exactly the same training strategy as introduced in Appendix C.1, except that for ReBAT++ we adopt PGD-12 with perturbation norm $\varepsilon_\infty = 10/255$ for training after LR decay. For Tiny-ImageNet, we follow the learning schedule of Chen et al. [8], in which the model is trained for a total of 100 epochs and the LR decays twice (by 0.1) at epoch 50 and 80.

**Choices of Hyperparameters.** For KD+SWA [8], PGD-AT+TE [12], AWP [53] and WA+CutMix [36], we strictly follow their original settings of hyperparameters. For MLCAT$_{WP}$ [54], we simply report the test results reported in their paper. Following Chen et al. [8], SWA/WA as well as the ReBAT regularization purposed in Section 4.1 start at epoch 105 (5 epochs later than the first LR decay where robust overfitting often begins), and following Rebuffi et al. [36] we choose the EMA decay rate of WA to be $\gamma = 0.999$. Please refer to Table 4 for our choices of the decay factor $d$ and regularization strength $\lambda$. We notice that since CutMix improves the difficulty of learning, the model demands a relatively larger decay factor to better fit the augmented data. For Tiny-ImageNet, we also apply a larger $\lambda$ after the second LR decay to better maintain the flatness of adversarial loss landscape and control robust overfitting. We provide more discussions on the choice of hyperparameters in Appendix D.1.

Table 4: Choices of hyperparameters when training models on different datasets using different network architectures with ReBAT.

| Network Architecture | Method | CIFAR-10/CIFAR-100 | Tiny-ImageNet |
|---|---|---|---|
| PreActResNet-18 | ReBAT | $d = 1.5, \lambda = 1.0$ | $d = 4.0, \lambda_1 = 2.0, \lambda_2 = 10.0$ |
| | ReBAT++ | $d = 1.7, \lambda = 1.0$ | $d = 4.0, \lambda_1 = 2.0, \lambda_2 = 10.0$ |
| | ReBAT+CutMix | $d = 4.0, \lambda = 2.0$ | $d = 6.0, \lambda = 1.5$ |
| WideResNet-34-10 | ReBAT | $d = 1.3, \lambda = 0.5$ | - |
| | ReBAT++ | $d = 1.3, \lambda = 0.5$ | - |
| | ReBAT+CutMix | $d = 4.0, \lambda = 2.0$ | - |

## C.5 Training Robust Accuracy

Figure 14a shows the robust accuracy change on the training data during the training. Compared with vanilla PGD-AT that yields training robust accuracy over 80% at epoch 200, ReBAT manages to suppress it to only 65%. It successfully prevents the trainer $\mathcal{T}$ from learning the non-robust features *w.r.t.* the training data too fast and too well, and therefore significantly reduces the robust generalization gap (from $\sim 35\%$ to $\sim 9\%$) and mitigates RO.

## C.6 Training Efficiency

We also test and report the training time (per epoch) of several methods evaluated in this paper. For a fair comparison, all the compared methods are integrated into a universal training framework and each test runs on a single NVIDIA GeForce RTX 3090 GPU.

From Table 5, we can see that ReBAT requires nearly no extra computational cost compared with vanilla PGD-AT in each epoch (136.2s *v.s.* 131.6s), implying that it is an efficient training method in practical. As we regularize the model trainer to restore the balance of the minimax game, we can see from Figure 14b that the training takes slightly longer to attain optimal performance, e.g., 110 epochs (PGD-AT) and 170 epochs (ours). However, in practice, people usually train longer in AT (typically 200 epochs) to attain better accuracy, and use early stopping to select the best checkpoint. As our

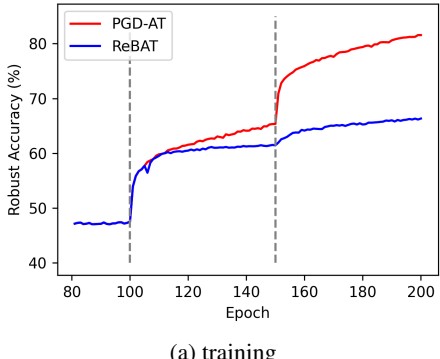
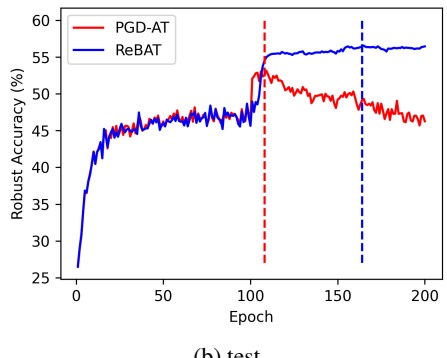

| (a) training | (b) test |
|---|---|

Figure 14: Training and test robust accuracy of PGD-AT and ReBAT on CIFAR-10 under the perturbation norm $\varepsilon_\infty = 8/255$ based on the PreActResNet-18 architecture. The gray dashed lines in (a) indicate LR decays, and the colored dashed lines in (b) indicate the epochs where each method attains its best robustness, respectively.

Table 5: Combining training time per epoch on CIFAR-10 under the perturbation norm $\varepsilon_\infty = 8/255$ based on the PreActResNet-18 architecture.

| Method | Training Time per Epoch (s) |
|---|---|
| PGD-AT | 131.6 |
| WA | 132.1 |
| KD+SWA | 131.6+16.5+141.7 |
| AWP | 142.8 |
| MLCAT$_{wp}$ | 353.3 |
| **ReBAT** | 136.2 |
| WA+CutMix | 168.6 |
| **ReBAT+CutMix** | 173.1 |

method does not need early stopping techniques (almost no robust overfitting), and have neglectable computation overhead per epoch, the total training time is comparable with vanilla AT. We also remark that KD+SWA, one of the most competitive methods that aims to address the RO issue, is not really computationally efficient as it requires to pretrain a robust classifier and a non-robust one as AT teacher and ST teacher respectively.

## D   More Experiments on ReBAT

In this section, we conduct extensional experiments on the proposed ReBAT method to further demonstrate its effectiveness, efficiency and flexibility.

### D.1   Additional Results on BoAT Loss

In Section C.4, we discuss the detailed configurations for the experiments in Figure 6a, where we show that BoAT can largely mitigate robust overfitting. Here, we further summarize the performance of best and final checkpoints of the original AT+WA method and our BoAT. As shown in Table 6, BoAT not only boosts the best robustness by a large margin (0.67% higher against AA) but also significantly suppresses RO (1.58% *v.s.* 6.06% against AA). To achieve the reported robustness, we first use $\lambda_1 = 10.0$ after the first LR decay and then apply $\lambda_2 = 60.0$ after the second LR decay to better maintain the flatness of adversarial loss landscape and control robust overfitting.

### D.2   Additional Results on the Effect of LR decay

Below we show that when a relatively large decay factor $d$ is applied, *i.e.,* the model has overly strong fitting ability that results in robust overfitting, a large regularization coefficient $\lambda$ should be chosen

Table 6: Comparing model robustness w/ and w/o BoAT loss on CIFAR-10 under the perturbation norm $\varepsilon_\infty = 8/255$ based on the PreActResNet-18 architecture.

| Method | Natural | | | PGD-20 | | | AutoAttack | | |
|---|---|---|---|---|---|---|---|---|---|
| | best | final | diff | best | final | diff | best | final | diff |
| AT+WA | **83.50** | **84.94** | -1.44 | 55.05 | 47.60 | 7.45 | 49.89 | 43.83 | 6.06 |
| ReBAT($d = 10$) | 81.54 | 82.42 | **-0.88** | **55.29** | **53.43** | **1.86** | **50.56** | **48.98** | **1.58** |

for better performance. Table 7 reveals this relationship between $d$ and $\lambda$. When $d = 1.3$, even a $\lambda$ as small as 1.0 will harm both the best and last robustness as well as natural accuracy, as $d = 1.3$ is already too small a decay factor that makes the model suffering from underfitting and naturally requiring no more flatness regularization. On the other side, when $d$ is relatively large, even a strong regularization of $\lambda = 4.0$ is not adequate to fully suppress RO. Besides, comparing the situation of $\lambda > 0$ and $\lambda = 0$ for a fixed $d(\geq 1.5)$, we emphasize that the purposed BoAT loss again exhibits its apparent effectiveness in simultaneously boosting the best robustness and mitigating RO.

Table 7: Changing decay factor $d$ and regularization strength $\lambda$ and evaluating model robustness on CIFAR-10 under the perturbation norm $\varepsilon_\infty = 8/255$ based on the PreActResNet-18 architecture.

| Method | Natural | | | PGD-20 | | | AutoAttack | | |
|---|---|---|---|---|---|---|---|---|---|
| | best | final | diff | best | final | diff | best | final | diff |
| ReBAT($d = 1.3, \lambda = 0.0$) | **81.17** | **81.27** | -0.10 | **56.43** | **56.23** | 0.20 | **50.80** | **50.75** | 0.05 |
| ReBAT($d = 1.3, \lambda = 1.0$) | 80.67 | 80.63 | 0.04 | 56.27 | 56.15 | 0.12 | 50.74 | 50.65 | 0.09 |
| ReBAT($d = 1.3, \lambda = 4.0$) | 78.50 | 78.36 | **0.14** | 55.10 | 55.04 | **0.06** | 50.31 | 50.30 | **0.01** |
| ReBAT($d = 1.5, \lambda = 0.0$) | **81.90** | **82.39** | -0.49 | 56.21 | 55.95 | 0.26 | 50.81 | 50.58 | 0.23 |
| ReBAT($d = 1.5, \lambda = 1.0$) | 81.86 | 81.91 | **-0.05** | **56.36** | **56.12** | **0.24** | **51.13** | **51.22** | -0.09 |
| ReBAT($d = 1.5, \lambda = 4.0$) | 79.68 | 79.88 | -0.20 | 55.72 | 55.65 | 0.07 | 50.52 | 50.50 | **0.02** |
| ReBAT($d = 4.0, \lambda = 0.0$) | **83.05** | **85.38** | -2.33 | 55.98 | 50.87 | 5.11 | 50.38 | 46.95 | 3.43 |
| ReBAT($d = 4.0, \lambda = 1.0$) | 82.46 | 84.84 | -2.38 | 55.86 | 52.02 | 3.84 | 50.87 | 47.88 | 2.99 |
| ReBAT($d = 4.0, \lambda = 4.0$) | 80.99 | 84.07 | -3.08 | **56.06** | **53.58** | 2.48 | **51.00** | **49.02** | 1.98 |

### D.3 Additional Results on Adopting Stronger Training Time Attacker

In Section 4.3, we design two versions of ReBAT, and we name the stronger one that also uses a stronger training attacker as ReBAT++. Here we fix the hyperparameter used in ReBAT above (note that we use a larger LR decay factor $d = 1.7$ in ReBAT++ for CIFAR-10, and now we still use $d = 1.5$ as in ReBAT) and adjust the attacker strength to study its influence when combined with ReBAT. According to Figure 15, though a slightly stronger attack (*e.g.*, $\varepsilon = 9/255$) may marginally improves the best and last robust accuracy, it heavily degrades natural accuracy, particularly when much stronger attack is used. We deem that this is because it breaks the balance from the other side that the overly strong attacker $\mathcal{A}$ dominates the adversarial game and results in an underfitting state that harms both robust and natural accuracy.

### D.4 Further Improving Natural Accuracy with Knowledge Distillation

Chen et al. [8] propose to adopt knowledge distillation (KD) [18] to mitigate RO and it is worth mentioning that their method achieves relatively good natural accuracy according to Table 1 and 2. Since our ReBAT method is orthogonal to KD, we propose to combine our techniques with KD to further improve natural accuracy. Specifically, we simplify their method by using only a non-robust standard classifier as a teacher (ST teacher) instead of using both a ST teacher and a AT teacher, because i) a large sum of computational cost for training the AT teacher will be saved, ii) our main goal is to improve natural accuracy so the ST teacher matters more and iii) ReBAT already use the WA model as a very good teacher. This gives the final loss function as

$$\ell_{\text{ReBAT+KD}}(x, y; \theta) = (1 - \lambda_{\text{ST}}) \cdot \ell_{\text{BoAT}}(x, y; \theta) + \lambda_{\text{ST}} \cdot \text{KL}\left(f_\theta(x) \| f_{\text{ST}}(x)\right), \quad (5)$$

where $f_{\text{ST}}$ indicates the ST teacher and $\lambda_{\text{ST}}$ is a trade-off parameter.

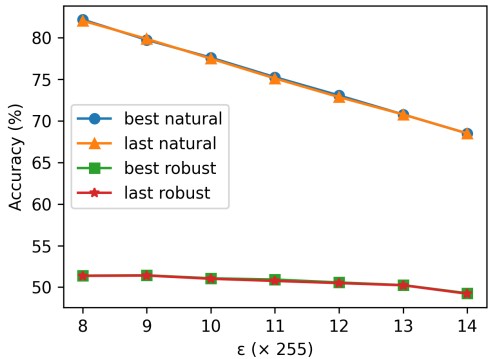

Figure 15: Using different adversarial attack strength in ReBAT.

Table 8: Combining our methods with knowledge distillation and evaluating model robustness on CIFAR-10 under the perturbation norm $\varepsilon_\infty = 8/255$ based on the PreActResNet-18 architecture.

| Method | Natural | | | PGD-20 | | | AutoAttack | | |
|---|---|---|---|---|---|---|---|---|---|
| | best | final | diff | best | final | diff | best | final | diff |
| ReBAT ($\lambda_{\mathrm{ST}} = 0.0$) | 81.86 | 81.91 | **-0.05** | **56.36** | **56.12** | 0.24 | **51.13** | **51.22** | -0.09 |
| ReBAT+KD ($\lambda_{\mathrm{ST}} = 0.4$) | 83.59 | 83.64 | **-0.05** | 54.91 | 54.77 | -0.14 | 50.70 | 50.99 | **-0.29** |
| ReBAT+KD ($\lambda_{\mathrm{ST}} = 0.5$) | 84.12 | 84.20 | -0.08 | 55.28 | 55.39 | -0.11 | 50.47 | 50.72 | -0.25 |
| ReBAT+KD ($\lambda_{\mathrm{ST}} = 0.6$) | **84.34** | **84.72** | -0.38 | 53.83 | 54.30 | **-0.47** | 50.15 | 50.37 | -0.22 |

Table 8 compares the performance of ReBAT+KD when different $\lambda_{\mathrm{ST}}$ is applied, and clearly a large $\lambda_{\mathrm{ST}}$ results in improvement in natural accuracy and decreases robustness (may due to the theoretically principled trade-off between natural accuracy and robustness [59]). However, it is still noteworthy that when $\lambda_{\mathrm{ST}} = 0.4$, a notable increase in natural accuracy ($\sim 1.7\%$) is achieved at the cost of only a small slide of $\sim 0.2\%$ in final robustness against AA. Also when $\lambda_{\mathrm{ST}} = 0.5$, it achieves comparable natural accuracy with KD+SWA [8] but has much higher AA robustness. Moreover, we emphasize that RO is almost completely eliminated regardless of the trade-off, which is the main concern of this paper and demonstrates the superiority of our method against previous ones. An intriguing phenomenon is that nearly all the final results are better than the results on the best checkpoints selected by the validation set, which implies that in this training scheme AT enjoys the same property of "training longer, generalize better" as ST without any need of early stopping.

### D.5  The Effect of Different Learning Rate Schedules

In the previous experiments we only investigate the piecewise LR decay schedule. However, a natural idea would be using mild LR decay schedules, *e.g.,* Cosine and Linear decay schedule, instead of suddenly decaying it by a factor of $d$ at some epoch in the piecewise decay schedule. As mentioned in Section 6, previous works have shown that changing LR decay schedule fails to effectively suppress RO whether with [45] or without WA [38] because the LR finally becomes small and endows the trainer with overly strong fitting ability. Therefore, here we continue to experiment with modified Cosine and Linear decay schedules that follow a similar LR scale of the piecewise LR decay schedule, and summarise the results in Table 9. To be specific, the LR still decays to 0.01 at epoch 150 and 0.001 at epoch 200, though the two decay stages (from epoch 100 to 150 and 150 to 200) are designed to be gradual following the Cosine/Linear schedule. We also gradually increase the strength of ReBAT regularization from zero as the LR gradually decreases, following the "larger decay factor goes with stronger ReBAT regularization" principle that we introduced in Appendix D.2.

It can be concluded from Table 9 that simply changing the LR decay schedule indeed improves the best robust accuracy from 49.89% to 50.59% and 50.67% against AA respectively, but it provides no help at all in mitigating RO as the final robust accuracy is still below 45% against AA. We also note that in this situation, the application of BoAT loss not only significantly mitigates RO but also further improves the best model robustness, which also proves its effectiveness.

Table 9: Comparing our method with WA on CIFAR-10 under the perturbation norm $\varepsilon_\infty = 8/255$ based on PreActResNet-18 architecture, when Cosine/Linear LR decay schedule are applied.

| Method | Cosine | | | | | | Linear | | | | | |
|---|---|---|---|---|---|---|---|---|---|---|---|---|
| | Natural | | PGD-20 | | AutoAttack | | Natural | | PGD-20 | | AutoAttack | |
| | best | final | best | final | best | final | best | final | best | final | best | final |
| WA(d=10) | **82.32** | **84.99** | 55.97 | 47.84 | 50.59 | 44.08 | **82.21** | **85.01** | 56.21 | 48.10 | 50.67 | 44.63 |
| **ReBAT(d=10)** | 82.06 | 82.22 | **56.12** | **54.44** | **50.98** | **49.81** | 81.98 | 82.13 | **56.33** | **54.50** | **51.08** | **49.63** |

## D.6 Results on Different Network Architectures

In previous experiments we compare methods based on PreActResNet-18 and WideResNet-34-10 architecture, and here we also adopt VGG-16 [41] and MobileNetV2 [39] architecture. The significant improvement against baseline PGD-AT in both best and final robust accuracy and in mitigating RO reported in Table 10 further demonstrates that our method works on a wide range of network architectures.

Table 10: Comparing our method with PGD-AT on CIFAR-10 under the perturbation norm $\varepsilon_\infty = 8/255$ based on VGG-16 and MobileNetV2 architecture.

| Method | VGG-16 | | | | | | MobileNetV2 | | | | | |
|---|---|---|---|---|---|---|---|---|---|---|---|---|
| | Natural | | PGD-20 | | AutoAttack | | Natural | | PGD-20 | | AutoAttack | |
| | best | final | best | final | best | final | best | final | best | final | best | final |
| PGD-AT | **78.43** | **81.64** | 50.56 | 44.25 | 44.19 | 39.66 | **79.85** | **80.67** | 51.56 | 50.67 | 46.01 | 45.27 |
| **ReBAT** | 78.17 | 78.37 | **53.13** | **53.01** | **47.24** | **47.13** | 78.98 | 80.81 | **53.18** | **52.57** | **47.66** | **47.35** |

