# OpenReview forum: "Balance, Imbalance, and Rebalance: Understanding Robust Overfitting from a Minimax Game Perspective"
_NeurIPS.cc/2023/Conference — NeurIPS 2023 poster_

### Official Review · Reviewer_kwEx · 2023-07-02

**Soundness:** 2 fair
**Presentation:** 3 good
**Contribution:** 2 fair
**Rating:** 4
**Confidence:** 4

**Summary:**

This paper explains the phenomenon of robust overfitting in adversarial training from a minimax game perspective. The author considers AT as a minimax game between the model trainer and the attacker, pointing out the imbalance between them leads to the network memorizing non-robust features, causing robust overfitting. Based on these explanations, the author proposes several measures to rebalance the minimax game, thereby mitigating robust overfitting and improving adversarial robustness.

**Strengths:**

1. It is interesting to study AT from a minimax game perspective.
2. The paper proposes multiple measures to alleviate robust overfitting and enhance adversarial robustness.

**Weaknesses:**

1. The motivation is unclear. The author's explanation of the robust overfitting process is based on some observation-driven analysis, which are difficult to be convincing. For example, the attacker injects non-robust features for misclassification, and the cause of robust overfitting is the network's memorization of non-robust features. What exactly are the false non-robust mapping and the falsely memorized non-robust features? Can the authors use the intuitive and precise statement to explain the mechanism of robust overfitting?

2. The method's novelty is limited. The author claims that previous attempts to change attacker strength have not focused much on robust overfitting. However, there is existing research in this area:
Yu C, Zhou D, Shen L, et al. Strength-Adaptive Adversarial Training, arXiv preprint arXiv:2210.01288, 2022.

3. The experimental results are incomplete and not significant. 1) Did the author confirm that the robustness of MLCAT in Table 1 is lower than AWP? 2) The author introduces multiple measures to mitigate robust overfitting and reports their combined performance. However, what are the individual performances of each technique? Considering that even combining multiple existing techniques for robust overfitting mitigation can further improve robustness, it is necessary to report the experimental results of each individual technique applied to Standard AT.

**Questions:**

Please refer to comments in weakness.

---

> ### Author Rebuttal · Authors · 2023-08-09
>
> We thank Reviewer kwEx for your time and efforts in reviewing this work. Below, we address your main concerns of this work.
>
> ---
>
> **Q1.** What exactly are the false non-robust mapping and the falsely memorized non-robust features? Can the authors use the intuitive and precise statement to explain the mechanism of robust overfitting?
>
> **A1.**  To address your concerns, we add both intuitive illustrations and formal definitions below. We will append these discussions in the revision to make it clearer.
>
> **1) Intuitive Understandings**
>
> Following your suggestion, we plot an intuitive figure Fig A (in Rebuttal PDF) to illustrate each individual stage during the robust overfitting. Due to the space limit, please refer to A2 of Reviewer 55GS for detailed descriptions.
>
> **2) Precise Definitions**
>
> In this paper, we mainly follow Ilyas et al’s definitions of robust and non-robust features, and extract the features following the same procedure. Thus, “non-robust mapping” and “non-robust features” refer to the same thing, because Ilyas et al define **robust/non-robust features are defined as mappings** $f:X\to \mathbb{R}$  (Sec 2). The key difference is that they only extract features only from the final model, while we study features learned at different training stages.
>
> To state it precisely, a training configuration $t\in\mathcal{T}$ (LR, epoch, etc) specifies a hypothesis class of features/mappings that it can robustly fit on training data, i.e.,
>
> $H_t=${$f_\theta:X\to R \|$ $E_{x, y\sim D_{train}}\inf_{\|x-\bar x\|\leq\varepsilon}[y \cdot f_\theta(x)]>0$ $\ (f_{\theta}$ is robust on training data), and $\theta\in{\Theta}$ is attainable under the configuration $t$}.
>
> We say a non-robust feature $f$ is falsely memorized if $f\notin H_{before}, f\in H_{after}$, where $H_{before},H_{after}$ refer to the hypothesis classes before and after LR decay, respectively. In other words, the non-robust $f$ is falsely memorized by the model (as a robust feature) under the after-LR-decay trainer. This mismatch could happen because the trainer only sees training examples and it can overfit them under an imbalanced minimax game. Since this feature is essentially a non-robust feature (in the population sense), this feature still behaves non-robust on test data, and even introduce shortcuts to test-time attack (verified in Sec 3.2.2).
>
> **3) Empirical Evidence**
>
> Notably, our explanation of robust overfitting is further justified by four extensive experiments in Section 3.2.2, where **we extract non-robust features from models before and after LR decay following Ilyas et al.’s procedure**, and compare their influence on robustness (Verification 1), target-class information (Verification 2), class correlation (Verification 3), and class symmetry (Verification 4). The phenomena fully support our dynamic game perspective of AT.
>
> ---
>
> **Q2.** The method's novelty is limited. The author claims that previous attempts to change attacker strength have not focused much on robust overfitting. However, there is existing research in this area (Yu et al., SAAT).
>
> **A2.** Thank you for pointing out this recent work, and we will add it for discussion in revision. Like ours, SAAT also considered stronger attack as a way to counter robust overfitting. However, we remark that there are several key differences:
>
> - **Different perspectives.** The **main focus/contribution in our work is the minimax game understanding of robust overfitting (Sec 3)**. The three strategies are naturally motivated by our understanding to rebalance the minimax game. Instead, the design of SAAT is motivated directly by the influence of the perturbation budget on robustness disparity. Therefore, the two works clearly have different motivations and design principles.
> - **Different solutions.** Notably, stronger attack is **only one of the three strategies proposed in our work** (Sec 4), and it is **optional** and **NOT adopted in the final ReBAT method** (bootstrap+small decay) because it degrades clean accuracy a lot. Besides, ReBAT attains **very good accuracy and robustness *without stronger attack*, and it can still outperform AWP+SAAT** (e.g., clean 79.49 v.s. 81.86, AA 49.29 v.s. 51.22) with neglectable overfitting.
> - **Different strategies for adjusting attacker.** The two are also different in the proposed stronger attack. SAAT gradually increases $\varepsilon$ along training and adjusts adaptively for different samples. Instead, ours is much simpler and easy to use: **we only apply larger perturbations after LR decay to counter stronger trainer, and use a constant perturbation budget for all samples** (Sec 4.2)**.** As shown in Table B in the attached PDF, this simple strategy achieves comparable or even better performance than SAAT (78.17 v.s. 76.37 natural acc, 50.99 v.s. 48.86 best AA, 47.66 v.s. 47.17 last AA), and still brings slight better robustness when combined with other methods like AWP and ReBAT.
>
> Thus, according to these key differences, our method is still quite different from SAAT and provides new perspectives and solutions for understanding and alleviating robust overfitting.
>
> ---
>
> **Q3.** Did the author confirm that the robustness of MLCAT (Yu et al.) in Table 1 is lower than AWP?
>
> **A3.** Indeed, there is a transcription mistake in the WideResNet-34 results of MLCAT in Table 1. Thanks for pointing it out. We add the correct results in Table A (Rebuttal PDF) and will fix them in the revision.
>
> ---
>
> **Q4.** What are the individual performances of each technique?
>
> **A4.** We note that we have provided a detailed analysis of each technique in **Figure 5 (Sec 4)** and included the quantitative results in **Appendix C**. We further summarize the results in Table C in Rebuttal PDF. We can see that each technique is useful for mitigating RO and combining them leads to better performance.
>
> ---
>
> Hope our elaborations and new results above could address your concerns. Please let us know if there is more to clarify.

---

> ### Author Response · Authors · 2023-08-14
> **Could you please have a look at our rebuttal?**
>
> Dear Reviewer kwEx, thanks for your time reviewing our paper. We have meticulously prepared a detailed response addressing the concerns you raised. Could you please have a look to see if there are further questions? Your invaluable input is greatly appreciated. Thank you once again, and we hope you have a wonderful day!

---

> ### Comment · Senior_Area_Chairs · 2023-08-20
> **Please take a look at author response and let us know if your opinion has changed.**
>
> Thank you.

---

> > ### Comment · Reviewer_kwEx · 2023-08-21
> >
> > I thank the authors for their response. However, I remain unconvinced by the explanation provided for the mechanism behind robust overfitting:
> > 1. If adversarial attacks are intended to introduce new non-robust features, why doesn't the network learn the easily learnable non-robust features from the dataset, similar to how it happens in standard training?
> > 2. Why does the model need to memorize non-robust features? Because these are non-robust features, why wouldn't attackers try to counteract them rather than letting the model memorize them?

---

> > > ### Author Response · Authors · 2023-08-21
> > > **Further Response to Reviewer kwEx**
> > >
> > > Thanks for your reply! We will address your remaining concerns below:
> > >
> > > ---
> > >
> > > **Q1.** If adversarial attacks are intended to introduce new non-robust features, why doesn't the network learn the easily learnable non-robust features from the dataset, similar to how it happens in standard training (ST)?
> > >
> > > **A1.** In our analysis, we follow the well-known robust/non-robust feature framework for understanding adversarial training [1,2], where the attacker creates misclassification by perturbing non-robust features, and the model tries to eliminate them. Built upon this framework, our minimax game perspective gives **a more practical characterization of AT dynamics**. Ideally, if a classifier only learns robust features during AT (as analyzed in [1]), then it will be perfectly robust (i.e., clean acc = robust acc). However, this does not hold in practice, since we always observe a large gap between accuracy and robustness. Thus, **practical AT models indeed learn useful but non-robust features during training**. And as a result, the attacker can use these non-robust features to generate misclassified adversarial samples, as we always observed in practice.
> > >
> > > On the other side, unlike ST, **learning more non-robust features will face countermeasures by the attacker in AT, so the trainer cannot memorize non-robust features as easily as ST**. For example, for a non-robust feature $f(x)$ assigned to class $y$, the attacker can add this feature to another example $x’$ from the $y’(\neq y)$ class to misclassify it, and the new adversarial example $(x’_{adv},y’)$ contains the feature $f(x)$ but it is now assigned to class $y’$. If the model further memorizes this new example, it will destroy the original $f(x)\to y$ feature (and this new feature will also be destoried in the next update). Thus, due to this adversarial effect, **the model will not be able to memorize all non-robust features generated by the attacker.**
> > >
> > > Notably, if the attacker and the model trainer are properly balanced (e.g., befor LR decay), in the long run, the two players can strick a balance, where the model can memorize no more non-robust features generated by the attacker, **as we analyzed and verified in** **Sec 3.1**. In this case, the model is not perfectly robust and the attacker can also generate adversarial examples, yet the two still maintain a roughly constant level of robustness (Fig 1a).
> > >
> > > For more empirical evidence, we refer to the discussions in Sec 3.1. We will add these explanations in the revision for better understanding.
> > >
> > > **Reference:**
> > >
> > > [1] Tsipras, et al. Robustness may be at odds with accuracy. ICLR. 2019.
> > >
> > > [2] Ilyas et al. Adversarial examples are not bugs, they are features. NeurIPS 2019.
> > >
> > > ---
> > >
> > > **Q2.** Why does the model need to memorize non-robust features? Because these are non-robust features, why wouldn't attackers try to counteract them rather than letting the model memorize them?
> > >
> > > **A2.** We will address your questions point by point.
> > >
> > > > Why does the model need to memorize non-robust features?
> > > >
> > >
> > > As the model trainer’s objective is $\min_\theta\ell_{\rm CE}(x_{\rm adv},y),$ its goal is to increase training accuracy on adversarial examples $x_{\rm adv}$. As non-robust features contained in $x_{adv}$ are useful for classification and thus help decrease training loss, **the model itself wants to memorize non-robust features whenever possible**. However, due to the countermeasures of the attacker, **how much this goal can be fulfilled depends on the relative strength between the attacker and the trainer**.
> > >
> > > > Because these are non-robust features, why wouldn't attackers try to counteract them rather than letting the model memorize them?
> > > >
> > >
> > > As we elaborated in **A1** above, if the trainer and the attacker are balanced, the model is unable to memorize all non-robust features because the attacker can constantly craft countermeasures.
> > >
> > > However, as we analyzed in Sec 3.2, the trainer and the attacker become imbalanced after LR decay, and endowed with smaller LR, **the stronger trainer can now draw a more complex region that robustly memorizes this non-robust feature on training set**. In this case, **the relatively weak attacker can no longer craft enough countermeasure.** As a result, the model memorizes more and more non-robust features and the attacker cannot counteract them, **which is directly revealed in the dramatic increase of training robust accuracy after LR decay (Fig 1)**. Besides, we have verified how this imbalance leads to the memorization of non-robust features in **Sec 3.1 (Fig 2a)**, and we further analyze how this memorization leads to RO with extensive experiments in **Sec 3.2.**
> > >
> > > ---
> > >
> > > Hope the explanations above could address your concerns. Please let us know if there is more to clarify. We are happy to take your further questions before the discussion stage ends!

---

> > > > ### Comment · Reviewer_kwEx · 2023-08-21
> > > >
> > > > Thank you for the author's prompt reply.
> > > >
> > > > Just as the author mentioned, for a non-robust feature $f(x)$ assigned to class y, if the attacker adds this feature to x' which does not belong to class y, then for the adversarial example $(x',y')$, why doesn't the model directly learn the non-robust features in x’ that are assigned to class y'?
> > > >
> > > > What are memorized non-robust features and non-memorized non-robust features? In my opinion, according to the definition of non-robust features in Line 93, under adversarial attacks, the model shouldn't have any intention to memorize these non-robust features. This is because even if they are memorized, the attacker still has the capability to completely destroy the correlations of these non-robust features.

---

> > > > > ### Author Response · Authors · 2023-08-21
> > > > > **AT models do contain non-robust features**
> > > > >
> > > > > Thanks for your prompt reply! We will address your further questions below.
> > > > >
> > > > > ---
> > > > >
> > > > > **Q1**. Just as the author mentioned, for a non-robust feature $f(x)$ assigned to class $y$, if the attacker adds this feature to $x'$ which does not belong to class $y$, then for the adversarial example $(x’,y’)$, why doesn't the model directly learn the non-robust features in x’ that are assigned to class y'?
> > > > >
> > > > > **A1**. We note that in AT, we use the adversarial example $(x^\prime_{adv}, y')$ for training, instead of the original sample $(x',y')$. In order to craft an adversarial example $x^\prime_{adv}$ that is misclassified to class $y$, the attacker will modify the original non-robust features in $x'$ (belonging to class $y'$) to $f(x)$ that belongs to class $y$. Thus, the fact that $x_{adv}$ is misclassified to $y$ (a successful attack) indicates that its contained non-robust features now belong to class $y$, which leads to the adversarial effect that we elaborated on in the last reply.
> > > > >
> > > > > ---
> > > > >
> > > > > **Q2**. What are memorized non-robust features and non-memorized non-robust features? In my opinion, according to the definition of non-robust features in Line 93, under adversarial attacks, the model shouldn't have any intention to memorize these non-robust features. This is because even if they are memorized, the attacker still has the capability to completely destroy the correlations of these non-robust features.
> > > > >
> > > > > **A2**. We respectfully disagree that the AT model does not memorize any non-robust features. In the **A1 of our last reply**, we explained the fact that although ideally, AT will only learn robust features, **practical AT models indeed learn some non-robust features**, in evidence of the gap between accuracy and robustness, and the fact that adversarial attacker can always craft successful attacks **(none of these will hold if the model does not memorize any non-robust features**)**.**
> > > > >
> > > > >  From our perspective, this discrepancy is caused by the limited capacity of the model on fitting all non-robust features. Indeed, before LR decay, due to the existence of non-robust features in AT models, the attacker can constantly craft a certain ratio of misclassified adversarial examples (about 30%). Yet, the training robust accuracy hinges on a constant level, indicating that these newly crafted non-robust features cannot be further fitted (or corrected) under the current trainer. Thus, there are some non-robust features memorized and some non-memorized.
> > > > >
> > > > > When the training becomes imbalanced after LR decay, the trainer with smaller LR becomes more capable at fitting non-robust features, while **the relatively weak attacker cannot craft enough countermeasures to destroy this correlation as before**. As a result, more and more non-robust features are memorized, leading to higher and higher training robust accuracy (Fig 1a). Importantly, we note that a feature that is **non-robust in the population sense** (the definition in Line 93) does NOT mean that it cannot be memorized robustly on the **finite training set**. When robust overfitting happens, the training robust accuracy can attain nearly 100%, but still, not all features it fits are truly robust on the test set. We have given a formal definition of these falsely memorized non-robust features in our **initial rebuttal (A1 (2))**.
> > > > >
> > > > > To summarize, the key message of our understanding above is that **despite the ideal settings that AT that won’t memorize any non-robust feature, we should focus more on how AT really behaves in practice**. And our analysis reveals that due to limited model capacity,  **real-world AT models indeed memorize some non-robust features and the degree of this memorization changes over the training process, dependent on the relative strength of the attack and the trainer**.
> > > > >
> > > > > ---
> > > > >
> > > > > Hope our elaborations above address your concerns. We are happy to take your further questions!

---

> > > > > > ### Comment · Reviewer_kwEx · 2023-08-21
> > > > > >
> > > > > > My intention is not to say that the model doesn't learn any non-robust features. I agree that the model may learn some non-robust features, leading to differences in natural accuracy and robust accuracy. I just didn't understand the difference between learn and memorize. So, when the paper refers to "memorize non-robust features," it's implying that the model learns non-robust features, correct? The features that the model learns become "memorized non-robust features," while those that are not learned are "non-memorized non-robust features," is that right?
> > > > > >
> > > > > > Considering the difference between natural accuracy and robust accuracy, it's evident that the model has been continuously learning non-robust features. Why didn't we observe any signs of robustness degradation before the change in learning rate? What triggers the degradation of robustness?
> > > > > >
> > > > > > Based on your responses, it seems that the triggering mechanism is the relative strength of the attack and the trainer. Then how can we explain the classical approach in AT where the additional dataset is used to mitigate robust overfitting? In this scenario, where the attacker remains the same and the model's robustness further improves, why isn't there robust overfitting instead?

---

> > > > > > ### Comment · Reviewer_kwEx · 2023-08-21
> > > > > >
> > > > > > In addition, for Q1, the inner maximization objective of AT is untargeted attack, not targeted attack from y' to y. How would the attacker know which specific class it misclassified into?

---

> > > > > > > ### Author Response · Authors · 2023-08-21
> > > > > > > **Thanks for your reply**
> > > > > > >
> > > > > > > Thanks for your prompt reply and clarifications!  We will address your remaining concerns as follows.
> > > > > > >
> > > > > > > ---
> > > > > > >
> > > > > > > **Q1**. My intention is not to say that the model doesn't learn any non-robust features. […] I just didn't understand the difference between learn and memorize.
> > > > > > >
> > > > > > > **A1**. Sorry for the confusion! We now get your point. Indeed, here “memorize” and “learn” mean the same thing, and we use them interchangeably most time. Sometimes we prefer to use the word “memorize” when the fitted non-robust features are not really generalizable to test data (particularly after robust overfitting). We will revise the paper to avoid potential confusion here.
> > > > > > >
> > > > > > > ---
> > > > > > >
> > > > > > > **Q2**. Considering the difference between natural accuracy and robust accuracy, it's evident that the model has been continuously learning non-robust features. Why didn't we observe any signs of robustness degradation before the change in learning rate? What triggers the degradation of robustness?
> > > > > > >
> > > > > > > **A2**. At the initial training stage, AT models can learn many new robust features, so their robustness keeps growing; although it also fits some non-robust features, overall robust features dominate the growth. Afterwards, we can observe that **model robustness actually saturates for a long time** (50-100 epochs). This is because after reaching a certain level of robustness, 1) the trainer cannot fit more robust features due to limited fitting ability; 2) it cannot fit more non-robust features either because, in view of its current fitting ability, the attacker is strong enough to counter it if it fits more non-robust features. Therefore, the two players reach constant robustness when they are properly balanced.
> > > > > > >
> > > > > > > **The key change induced by LR decay is the broken balance between the fitting and attacking abilities of the two players.** Specifically, after LR decay, the trainer with a smaller LR can draw a more complex decision boundary around training samples to fit the non-robust features, while the static attacker can no longer trigger them out. As **this imbalance persists, the trainer captures more and more remaining non-robust features and it is unstoppable by the weak trainer,** which leads to the continuous robust degradation after RO.
> > > > > > >
> > > > > > > ---
> > > > > > >
> > > > > > > **Q3**. Based on your responses, it seems that the triggering mechanism is the relative strength of the attack and the trainer. Then how can we explain the classical approach in AT where the additional dataset is used to mitigate robust overfitting? In this scenario, where the attacker remains the same and the model's robustness further improves, why isn't there robust overfitting instead?
> > > > > > >
> > > > > > > **A3**. This is an interesting question! In Sec 5, we revisited existing techniques to mitigate RO, and explained them from our perspective. For the data regularization part, we explained how data augmentations can reduce RO by corrupting non-robust features. **Adding training data, on the other hand, will significantly increase the difficulty for the model to robustly memorize the non-robust features.**
> > > > > > >
> > > > > > > As these features are essentially non-robust (in the population sense), adding more data will further approach the population distribution. Particularly, the model memorizing the original training data is still non-robust on the new data, so it needs to further spare more of its fitting abilities to these new data to robustly memorize the same non-robust feature. As a result, **using a larger dataset will make it harder for the trainer to memorize the non-robust features on so many samples**. So, using a larger dataset also helps prevent RO.
> > > > > > >
> > > > > > > Thanks for your insight, we will append this discussion to the “data regularization” part in Sec 5.
> > > > > > >
> > > > > > > ---
> > > > > > >
> > > > > > > **Q4**. In addition, for Q1, the inner maximization objective of AT is untargeted attack, not targeted attack from y' to y. How would the attacker know which specific class it misclassified into?
> > > > > > >
> > > > > > > **A4**. Indeed, AT uses untargeted attack. In this example, $y$ is not necessarily given *a priori;* instead, **it can also denote any class that $x^\prime_{adv}$ (e.g., crafted by the untargeted attacker) is misclassified into**. The fact that $x^\prime_{adv}$ is misclassified to $y$ indicates that $x^\prime_{adv}$ contains non-robust features (exacted from the model) w.r.t. class $y$. And in the same vein, fitting $(x'_{adv},y’)$ still triggers the adversarial effect on the $y$-class non-robust features in the model. So our analysis of the adversarial effect still holds for untargeted attack.
> > > > > > >
> > > > > > > ---
> > > > > > >
> > > > > > > Hope the clarification above could address your concerns. Please let us know if there is more to clarify.

---

> > > > > > > > ### Comment · Reviewer_kwEx · 2023-08-21
> > > > > > > >
> > > > > > > > If the model learns more non-robust features after the learning rate decreases, why is it that the gap between training robust accuracy and training standard accuracy becomes smaller after the learning rate decay?

---

> > > > > > > > > ### Author Response · Authors · 2023-08-21
> > > > > > > > > **Would you consider reevaluating the paper?**
> > > > > > > > >
> > > > > > > > > This phenomenon actually aligns well with our theory and is explained in Sec 3.2 (L126-131). After LR decay, the stronger trainer memorizes the non-robust features in the adversarial examples and makes these samples no longer attackable, which pushes up the overall training robust accuracy dramatically, as we have shown in Fig 1a (our paper). Since the training standard accuracy is already very high at that time (around 90%), the large increase in training robust accuracy (45%→80%) induces a decrease in their gap. As another piece of evidence, Fig 1 from Rice et al [1] gives a clear illustration of this process.
> > > > > > > > >
> > > > > > > > > We are always happy to take your further questions! Meanwhile, given that many of your previous concerns seem to have been resolved, would you please consider reevaluating our paper?
> > > > > > > > >
> > > > > > > > > **Reference:**
> > > > > > > > >
> > > > > > > > > [1] Rice et al. Overfitting in adversarially robust deep learning. ICML 2020

---

> > > > > > > > > > ### Comment · Reviewer_kwEx · 2023-08-22
> > > > > > > > > > **Response to would you consider reevaluating the paper**
> > > > > > > > > >
> > > > > > > > > > Thanks for the clarification. The experiments conducted in this work are solid. However, the novelty and explanation regarding robust overfitting is limited and not convincing, as outlined below. Therefore, I raised my score to 4.
> > > > > > > > > >
> > > > > > > > > > 1. The viewpoint of the relative strength of the attacker and the trainer as the trigger mechanism for robust overfitting is not convincing, which can not adequately explain various empirical behaviors within robust overfitting, such as the method of additional training data. Besides, there is no rigorous proof showing the relationship between non-robust features and robust overfitting.
> > > > > > > > > >
> > > > > > > > > > 2. Lacking novelty, despite wrapping the explanation of robust overfitting in the minimax game perspective, similar viewpoints have been proposed and discussed in prior works.

---

> > > > > > > > > > > ### Author Response · Authors · 2023-08-22
> > > > > > > > > > > **Thanks for your reply and further questions**
> > > > > > > > > > >
> > > > > > > > > > > Thank you for appreciating our response and the solidness of our experiments. However, we do not fully understand your remaining concerns, and respectfully hope that you could concretely explain your remaining concerns.
> > > > > > > > > > >
> > > > > > > > > > > > The viewpoint of the relative strength of the attacker and the trainer as the trigger mechanism for robust overfitting is not convincing, which can not adequately explain various empirical behaviors within robust overfitting, such as the method of additional training data.
> > > > > > > > > > > >
> > > > > > > > > > >
> > > > > > > > > > > We have provided a detailed explanation of how our understandings can naturally explain the methods using additional data in [A3 of our 4th-round reply](https://openreview.net/forum?id=yBoVwpGa5E&noteId=L47RaoTRc0). Could you please let us know which part of it you find problematic?
> > > > > > > > > > >
> > > > > > > > > > > For several other phenomena you mentioned, we have also provided natural and clear explanations with our framework. We even show that many existing techniques for alleviating RO can be understood within our framework in Sec 5. Please let us know if there is more to clarify.
> > > > > > > > > > >
> > > > > > > > > > > > Lacking novelty, despite wrapping the explanation of robust overfitting in the minimax game perspective, similar viewpoints have been proposed and discussed in prior works.
> > > > > > > > > > >
> > > > > > > > > > > We have extensively elaborated the difference to the previous work you mentioned, SAAT, in [our rebuttal](https://openreview.net/forum?id=yBoVwpGa5E&noteId=lf1h6AxFIT). As far as we can see, **SAAT does not explain robust overfitting from the minimax game perspective**. Specifically, our analysis provides a holistic understanding of robust overfitting based on the robust/non-robust feature framework, and verified this view with extensive experiments in Sec 3.2.2; and in our analyis, we mainly consider the RO induced by the trainer’s LR decay behaviors. In comparison, SAAT 1) only empirically studies the influence of attacker strength, 2) does not propose or adopt a minimax game view; 3) does not explain robust overfitting from robust/non-robust features. Thus, our work provides new insights on many aspects, and the designed method ReBAT outperforms SAAT significantly without using any similar techniques.
> > > > > > > > > > >
> > > > > > > > > > > If there is other related work that also explains robust overfitting from a minimax game perspective like ours, please let us know.
> > > > > > > > > > >
> > > > > > > > > > > > Besides, there is no rigorous proof showing the relationship between non-robust features and robust overfitting.
> > > > > > > > > > >
> > > > > > > > > > > As for your remaining concern about the lack of theoretical proof, we note that this is mainly because RO is a complex phenomenon of deep NNs and it is hard to be realized with theoretically tractable toy models. As far as we know, existing explanations of RO mostly rely on empirical justifications and do not have theoretical proof, including [1].  In view of this situation, we also designed extensive verification of our explanations and they all agree well with our analysis. Thus, we believe that our analysis from a minimax game perspective using robust/non-robust features can further the understanding of robust overfitting.
> > > > > > > > > > >
> > > > > > > > > > > We are looking forward to more concrete explanations of the above limitations you mentioned.
> > > > > > > > > > >
> > > > > > > > > > > **Reference:**
> > > > > > > > > > >
> > > > > > > > > > > [1] Yu et al. Understanding Robust Overfitting of Adversarial Training and Beyond. ICML 2022.

---

### Official Review · Reviewer_EtSp · 2023-07-03

**Soundness:** 3 good
**Presentation:** 3 good
**Contribution:** 3 good
**Rating:** 6
**Confidence:** 4

**Summary:**

This paper empirically shows that robust overfitting is caused by the over-memorization of the non-robust features after learning rate decay. To mitigate the issue of robust overfitting, the authors propose to use a stronger training attack, a smaller learning rate decay rate, and a bootstrapped adversarial training loss. The comprehensive empirical results validate the effectiveness of the proposed method in mitigating robust overfitting and even improving robustness.

**Strengths:**

1. This paper provides comprehensive verifications for their proposed reason for robust overfitting. The authors clearly show the effect of the false memorization of the adversarial non-robust features after learning decay in robust overfitting.

2. The empirical experiments on various datasets and networks are comprehensive. The results support the authors’ claim.


**Weaknesses:**

1. Besides extensive empirical results, it would be better for the authors to provide some analyses from a theoretical perspective (possibly using game theory which could be related to the empirical results in this paper).

2. The proposed method could hurt the natural test accuracy to some extent.


**Questions:**

1. What is the effect of the hyper-parameter \lambda?

2. Could you provide the result evaluated on CIFAR-10 in the experiments part? I am afraid that I did not find the corresponding results.


**Limitations:**

The proposed method can effectively mitigate the issue of robust overfitting.

---

> ### Author Rebuttal · Authors · 2023-08-09
>
> We thank Reviewer EtSp for appreciating the comprehensiveness and solidness of the verification of our understanding. Below, we address your main concerns.
>
> ---
>
> **Q1.** It would be better for the authors to provide some analyses from a theoretical perspective (possibly using game theory which could be related to the empirical results in this paper).
>
> **A1.** There is a body of literature that analyzes AT from a game theory perspective, see [1-5]. However, existing theory papers only analyze AT’s Nash equilibrium under toy models (e.g., Gaussian features and linear models), and **static** minimax players. **None of the existing literature considers a dynamic game and could explain robust overfitting from a game perspective, particularly in practical AT algorithms, as done in our work.** Nevertheless, we find some interesting connections and new insights between them that worth noting.
>
> **Connections.** Under these simplified assumptions, prior works show that although the Nash equilibrium of the AT objective exists and is robust, **the current alternating optimization of AT may fail to converge to the Nash equilibrium**, see, e.g., a recent work [5]. The key reason is that the trainer can falsely fit the non-robust features [5], which is in a similar spirit to our analysis that robust overfitting is caused by the falsely memorized non-robust features after LR decay. **This shows that our explanation is not only in line with the cutting-edge theory of AT, but also further explains robust overfitting from a game perspective for the first time.**
>
> **New insights.** We also notice that existing AT game theory papers cannot explain robust overfitting, and from the perspective of our theory, this is possible because they overlook the dynamic change of the AT players. Thus, our understanding of AT from a dynamic game can inspire more in-depth theoretical characterization of AT from the game theory perspective.
>
> As our work mainly focuses on the understanding robust overfitting of practical AT algorithms, and it is generally hard to theoretically analyze the training dynamics of practical DNNs, we leave more theoretical investigations under strong assumptions to future work. Following your suggestion, we will add this discussion on the relation to game theory in the revision.
>
> **References:**
>
> [1] Bulo et al. Randomized prediction games for adversarial machine learning. TNNLS, 2016.
>
> [2] Pal & Vidal. A game theoretic analysis of additive adversarial attacks and defenses. NeurIPS, 2020.
>
> [3] Bose et al. Adversarial example games. NeurIPS, 2020.
>
> [4] Pinot et al. Randomization matters how to defend against strong adversarial attacks. ICML, 2020.
>
> [5] Balcan et al. Nash Equilibria and Pitfalls of Adversarial Training in Adversarial Robustness Games. AISTATS, 2023.
>
> ---
>
> **Q2.** The proposed method could hurt natural accuracy to some extent.
>
> **A2.** Due to the well-known trade-off between accuracy and robustness, better robustness often comes with lower accuracy. Nevertheless, Table 1 (quoted below) shows that our ReBAT version achieves **both better accuracy and robustness** **than PGD-AT, TRADES, and AWP** on CIFAR-10. The ReBAT[strong] version utilizes stronger attacker, so it scarifies accuracy to trade for higher robustness. **Table 8 (Appendix C.4)** shows that further incorporating the knowledge distillation (KD) technique can further boost the clean accuracy of ReBAT while maintaining high robustness. Therefore, ReBAT achieves a good trade-off between accuracy and robustness as it can attain good clean accuracy while significantly improve model robustness.
>
> *Performance of best-epoch models on CIFAR-10 with PreActResNet-18. (Quoted from Table 1)*
>
> | Method | Natural Accuracy | AutoAttack |
> | --- | --- | --- |
> | PGD-AT | 81.62 | 47.51 |
> | TRADES | 80.45 | 48.09 |
> | AWP | 81.11 | 50.09 |
> | ReBAT (ours) | 81.86 | 51.13 |
> | ReBAT[strong] (ours) | 78.71 | 51.49 |
> | ReBAT + KD  (ours) (Table 8) | 84.20 | 50.72 |
>
> ---
>
> **Q3.** Effect of the hyper-parameter $\lambda$.
>
> **A3.** Here, $\lambda$ controls the strength of the KL regularization, and a larger $\lambda$ imposes stronger regularization on model flatness. In **Table 7 in Appendix C.2**, we include a detailed analysis of the influence of $\lambda$. We quote some results below. We can see that under a small decay rate ($d=1.5$), a medium $\lambda$ attains the best performance, while too large $\lambda$ hurts both natural accuracy and robustness severely with too much flat regularization. When the decay rate is larger, e.g., $d=4$, a larger $\lambda$ can significantly improve model robustness by regularizing the model’s fitting ability, but meanwhile, its natural accuracy also degrades. In view of this analysis, in practice, we simply choose a small decay factor $d=1.5$ and a medium $\lambda=1.0$ as it can attain both good accuracy and robustness.
>
> *Analysis of the decay factor ($d$) and regularization strength ($\lambda$) on CIFAR-10 with PreActResNet-18 (best-epoch results). (Quoted from Table 7)*
>
> | $d$ | $\lambda$ | Natural Accuracy | AutoAttack |
> | --- | --- | --- | --- |
> | 1.5 | 0.0 | 81.90 | 50.81 |
> |  | 1.0 | 81.86 | 51.13 |
> |  | 4.0 | 79.68 | 50.52 |
> | 4.0 | 0.0 | 83.05 | 50.38 |
> |  | 1.0 | 82.46 | 50.87 |
> |  | 4.0 | 80.99 | 51.00 |
>
> ---
>
> **Q4.** Missing CIFAR-10 results in the experiments part.
>
> **A4.** This is caused by a typo. Table 1 actually is the CIFAR-10 results with two different backbones. The correct caption should be “Comparing our method with several training methods on CIFAR-10 under the perturbation norm $ε_\infty = 8/255$ based on the PreActResNet-18 and WideResNet-34-10 architectures.” We will correct it in the revision. Thanks for pointing it out.
>
> ---
>
> Hope our elaborations and new results above could address your concerns. Please let us know if there is more to clarify.

---

> > ### Comment · Reviewer_EtSp · 2023-08-14
> > **My concerns are solved**
> >
> > Thanks for your replies. It is impressive to see the proposed method can improve both natural and robust test accuracy. Besides, it seems that the proposed method provides some new insights regarding explaining robust overfitting, which would be beneficial to future research in this area. Therefore, I still lean toward Weak Acceptance.

---

### Official Review · Reviewer_55GS · 2023-07-05

**Soundness:** 3 good
**Presentation:** 2 fair
**Contribution:** 4 excellent
**Rating:** 7
**Confidence:** 4

**Summary:**

This paper studies the robust overfitting phenomenon in adversarial training. Meanwhile, this paper focuses on a specific problem “the robust overfitting occurs when we use learning rate decay techniques.” This paper proposes a game perspective to explain the robust overfitting. It claims that the robust overfitting happens because of the imbalance between the attacker and the model after the LR decay. Moreover, this paper proposes ReBAT method to improve the robustness and mitigates the overfitting phenomenon.

**Strengths:**

This paper studies a specific but interesting problem, the robust overfitting phenomenon. This paper explains that the overfitting happens because of the learned mapping of the non-robust features after LR decay.

This paper provides an intersting perspective to help us to understand the cause of the robust ovetfitting. In particular, the robust problem is a min-max problem, which could be regarded as a game. The robust overfitting is because of the breaks of the original equilibrium.

This paper provides extensive verification to support its explanation, which addresses most of my questions when I first review this paper.

**Weaknesses:**

1.For the Figure 2 (b), I guess the red line denotes  w/ LR decay and the blue line denotes w/o LR decay;

2.For section 3.2.1, the authors share an interesting and important insight. I’m afraid it is a little trifling. Could the author provide a figure to demonstrate it?

3.In section 4.1, the authors provide some techniques to address the robust overfitting, e.g., bootstrapping, small decay factor. From the theoretical analysis in this paper,  I guess there may exist other methods to achieve the balanced learning of robust and non-robust feature, and more discussions is helpful.  Could the author provide some theoretical analysis about the proposed method for the re-balance?

4.I find the proposed method ReBAT may cause more computation overhead to get a convergent result. Could the author discusses the limitations of this work?

5.Could the author share more discussions about the game idea?  For example, in multi-task learning, could it be regarded as a game, where there are multiple players?

**Questions:**

See weakness.

---

> ### Author Rebuttal · Authors · 2023-08-09
>
> We thank Reviewer 55GS for your careful reading and appreciation on the novelty and solidness of our work. Below, we address your main concerns of this work.
>
> ---
> **Q1.** For Figure 2 (b), I guess the red line denotes w/ LR decay and the blue line denotes w/o LR decay.
>
> **A1.** Thank you for pointing it out. You are correct. We will correct it in the revision.
>
> ---
> **Q2.** For section 3.2.1, the authors share an interesting and important insight. Could the author provide a figure to demonstrate it?
>
> **A2.** Following your suggestion, we plot an intuitive figure Fig A (Rebuttal PDF) to illustrate each individual stage during the robust overfitting. Specifically, Fig A (a) shows that before LR decay it achieves a balance between the trainer and the attacker, and the weak trainer will NOT fit the non-robust features in misclassified adversarial training examples (blue→orange). Fig (b) shows that after LR decay, the trainer endowed with stronger fitting ability memorizes the adversarial examples to their original labels (blue) by drawing a more complex decision boundary (from a feature view, the non-robust features are falsely memorized altogether). Fig (c) shows that this new decision boundary introduces easy-to-reach shortcuts to attack test examples as it falsely maps y’-class features (the orange region) to the y class. As a result, the training robustness rises but test robustness degrades. This is in line with our analysis in Sec 3.2.1 that after decay, the falsely memorized non-robust features open shortcuts for test-time attack and lead to robust overfitting.
>
> ---
> **Q3.** From the theoretical analysis in this paper, I guess there may exist other methods to achieve the balanced learning of robust and non-robust feature, and more discussions are helpful. Could the author provide some theoretical analysis about the proposed method for the re-balance?
>
> **A3.** Indeed, there are other kinds of strategies to restore the balance of minimax training, including some tricks proposed in prior works. We have provided an overview of these strategies from our theoretical perspective in **Section 5**, where we explain how existing techniques (10+) can help restore the balance from three different aspects: data regularization, training regularization, and stronger attacker.
>
> Here, we provide a more detailed explanation of the proposed methods (bootstrap, small decay, and stronger attack). As shown in the theory of Section 3.2, robust overfitting rises when the model trainer memorizes false non-robust features contained in adversarial examples. Therefore, there are two ways to alleviate this effect, either to regularize the trainer (cannot fit), or to strengthen the non-robust features (cannot be fitted). The bootstrap regularization enforces a flat landscape of model weights (simpler decision boundary), such that the non-robust features cannot be memorized via drawing complex decision boundaries. Similarly, the small LR decay factor regularizes the optimizer’s local fitting ability, which also prevents memorizing the non-robust features. Stronger attacker, on the other side, makes the non-robust features too strong to be memorized.
>
> Hope the explanations above helps address your questions and we will add them in the revision.
>
> ---
> **Q4.** I find the proposed method ReBAT may cause more computation overhead to get a convergent result.
>
> **A4.** To ease your concerns, we calculate the per-epoch training cost of different methods. From the table below, we can see that the per-epoch training cost of ReBAT is almost the same as vanilla AT (129.4s v.s. 125.3s), lower than many other advanced AT methods (e.g., AWP). Thus, in this view, ReBAT is rather computationally efficient.
>
> As you mentioned and can be observed from the training process in Figure B (Rebuttal PDF), ReBAT takes slightly longer training time to converge to the best performance (it peaks at around the 170th epoch while AT peaks at around the 110th epoch), because it adopts smaller LR decay and regularization to avoid robust overfitting. **But it achieves much higher robustness and does not suffer from robustness deterioration.** Further considering that in practice, people usually train longer in AT (typically 200 epochs) to attain better accuracy and use early stopping to select the best checkpoint, the two methods actually require similar total training epochs in practice.
>
> Therefore, we believe that the computation cost of ReBAT is comparable to vanilla AT while attaining much better robustness.
>
> *Training time comparison on CIFAR-10 with PreActResNet-18 backbone.*
>
> | Method | Training Cost (per epoch) |
> | --- | --- |
> | PGD-AT | 125.3 |
> | AWP | 135.6 |
> | ReBAT | 129.4 |
>
> ---
> **Q5.** Could the author share more discussions about the game idea?
>
> **A5.** There is a body of literature that analyzes AT using game theory, e.g., [1], usually regarding AT as a zero-sum game between two players. A recent work argues that AT should be treated as a non-zero-sum game [2]. For multi-objective AT, there could exist multiple attackers that constitute a game of multiple players, where the AT could behave very differently.
>
> Due to word limit, please refer to **A1 in our response to Reviewer EtSp** for deeper discussions on the connections between our work and those papers and new insights our work brings to the game idea. Existing game theory papers mainly study AT’s Nash equilibrium under toy models and **static minimax players**, but not **a dynamic game in practical AT algorithms** as in our understandings in robust overfitting. We will elaborate this part in the revision.
>
> **References:**
>
> [1] Balcan et al. Nash Equilibria and Pitfalls of Adversarial Training in Adversarial Robustness Games. AISTATS, 2023.
>
> [2] Robey et al. Adversarial Training Should Be Cast as a Non-Zero-Sum Game. *arXiv preprint arXiv:2306.11035.* 2023.
>
> ---
> Hope our elaborations and new results above could address your concerns. Please let us know if there is more to clarify.

---

> > ### Comment · Reviewer_55GS · 2023-08-19
> > **Reply**
> >
> > Thank you for the comprehensive explanation.
> >
> > Your clarification has effectively addressed my concerns. The game idea presents a compelling direction within adversarial robustness. I believe such insights could potentially influence developments in other areas as well. I'm inclined to adjust my score to a 7.

---

### Official Review · Reviewer_Znmp · 2023-07-06

**Soundness:** 3 good
**Presentation:** 3 good
**Contribution:** 3 good
**Rating:** 6
**Confidence:** 4

**Summary:**

This paper investigates the phenomenon of robust overfitting in adversarial training and explains it from a minimax game perspective. The authors analyze how the decay of the learning rate disrupts the balance between the model trainer and the attacker, leading to robust overfitting. They propose a method called ReBalanced Adversarial Training (ReBAT) to mitigate robust overfitting and achieve good robustness even after long training.

**Strengths:**

1 The paper is well organized and easy to follow.

2  The paper provides a holistic understanding of robust overfitting in adversarial training by analyzing the imbalance between the model trainer and the attacker from a minimax game perspective. This perspective helps explain why robust overfitting occurs in adversarial training and why it does not occur in other training methods.

3 The experiments cover different network architectures and benchmark datasets, providing strong empirical evidence.

**Weaknesses:**

1 it is better to clearly state the diffference between the defined robust/non-robust features and the previous one in [15]

2 ReBAT[strong] seems like a strange notation, please consider to change one.

3 The caption of Table 1 seems not correct?

4 Does the minimax game view provide any insights on the accuracy and robustness tradeoff?

**Questions:**

Please see the above weaknesses.

**Limitations:**

Missing...

Please stately explicitly the limitations and potential negative impact.

---

> ### Author Rebuttal · Authors · 2023-08-09
>
> We thank Reviewer Znmp for your careful reading and for appreciating the proposed understanding and method for robust overfitting. Below, we address your main concerns of this work.
>
> ---
>
> **Q1.** Diffference between the defined robust/non-robust features and the previous one in Ilyas et al.
>
> **A1.** In this paper, we **follow Ilyas et al’s definitions of robust and non-robust features**, and extract the features following the same procedure. The main difference is that, instead of focusing only on the last trained model (as in Ilyas et al), we study **the dynamic behaviors of AT models on utilizing these robust/non-robust features during the training process.** Specifically, we extract non-robust features from models obtained before and after LR decay, and use these features to verify our understanding of robust overfitting from a dynamic minimax game perspective (Sec 3.2).
>
> ---
>
> **Q2.** ReBAT[strong] seems like a strange notation, please consider to change one.
>
> **A2.** Here, ReBAT[strong] refers to the combination of ReBAT with stronger attack. **ReBAT++** could be a good alternative name since this version further enhances adversarial robustness over ReBAT. Please let us know if you have better options.
>
> ---
>
> **Q3.** The caption of Table 1 seems not correct?
>
> **A3.** Thanks for pointing it out. There is a typo here. Table 1 shows CIFAR-10 results with different backbones. We will change the caption to “Comparing our method with several training methods on CIFAR-10 under the perturbation norm $ε_\infty = 8/255$ based on the PreActResNet-18 and WideResNet-34-10 architectures.”
>
> ---
>
> **Q4.** Insights on the accuracy and robustness trade-off from our view of minimax game.
>
> **A4.** The minimax view can also provide new insights into understanding the accuracy-robustness trade-off. As we discussed in the paper, during AT, $\mathcal{A}$ attacks by injecting non-robust features into adversarial examples.
> - When the minimax game is **balanced**, these non-robust features will not be fitted (i.e., discarded). Since these non-robust features also contribute to classification accuracy (see Ilyas et al.), discarding more non-robust features (i.e., more robust) will lead to worse accuracy.
> - When the minimax is **imbalanced** and the model trainer is stronger than the attacker (as studied in our paper), the model will capture more and more non-robust features and get better accuracy, while at the cost of introducing shortcuts to the attacker and leading to worse and worse robustness (revealed in Section 3.2).
>
> Therefore, in both balanced and imbalanced minimax game, there will be a trade-off between accuracy and robustness, and the discrepancy is larger when robust overfitting happens.
>
> ---
>
> **Q5.** Please state explicitly the limitations and potential negative impact.
>
> **A5.** Thanks for your suggestion. A main limitation of the proposed method is that as we regularize the model trainer to restore the balance of the minimax game, the training takes slightly longer to attain optimal performance, e.g., 110 epochs (AT) and 170 epochs (ours). However, in practice, people usually train longer in AT (typically 200 epochs) to attain better accuracy, and use early stopping to select the best checkpoint. As our method does not need early stopping techniques (without robust overfitting), and have neglectable computation overhead per epoch, the total training time is comparable with vanilla AT.
>
> As for the social impact, since our method is designed for better defence of adversarial attack, it should have a positive impact by enhancing the security of machine learning models.
>
> ---
>
> Hope our elaborations above could address your concerns. Please let us know if there is more to clarify.

---

### Author Rebuttal · Authors · 2023-08-09

The Rebuttal PDF can be seen in the attached file, which contains
- Figure A: an intuitive illustration for the proposed understanding of robust overfitting;
- Figure B: a plot of the training process comparing ReBAT with vanilla AT;
- Table A, B, C: additional comparison experiments between different AT methods.

---

### Decision · Program_Chairs · 2023-09-21

**Decision:**

Accept (poster)

**Comment:**

The reviewers have generally agreed that this paper provides a valuable contribution to the understanding of robust overfitting in adversarial training. The authors' approach of analyzing the issue from a minimax game perspective is appreciated for its novelty and the insights it provides. The proposed ReBalanced Adversarial Training (ReBAT) method is also seen as a promising solution to mitigate robust overfitting.

However, the reviewers have also pointed out some areas that could be improved. These include the need for clearer explanations and notations, more theoretical analysis, and a discussion on the limitations of the proposed method. The authors are encouraged to address these points in their final version of the paper.

Overall, the paper is accepted for its technical solidity, potential impact, and comprehensive empirical results. The authors are commended for their work and encouraged to continue their research in this area.